# Treatment, toxicity, and mortality after subsequent breast cancer in female survivors of childhood cancer

Cindy Im[1], Hasibul Hasan[1], Emily Stene[1], Sarah Monick[2], Ryan K. Rader[3], Jori Sheade[4], Heather Wolfe[5], Zhanni Lu[1], Logan G. Spector [1], Aaron J. McDonald[6], Vikki Nolan[6], Michael A. Arnold[7], Miriam R. Conces[8], Chaya S. Moskowitz[9], Tara O. Henderson[10], Leslie L. Robison[6], Gregory T. Armstrong[6], Yutaka Yasui [6], Rita Nanda[11], Kevin C. Oeffinger [12], Joseph P. Neglia[1], Anne Blaes [13] & Lucie M. Turcotte [1] ✉

Childhood cancer survivors, particularly those who received chest radiotherapy, are at high risk for developing subsequent breast cancer. Minimizing long-term toxicity risks associated with additional radiotherapy and chemotherapy is a priority, but therapeutic tradeoffs have not been comprehensively characterized and their impact on survival is unknown. In this study, 431 female childhood cancer survivors with subsequent breast cancer from a multicenter retrospective cohort study were evaluated. Compared with one-to-one matched females with first primary breast cancer, survivors are as likely to be prescribed guideline-concordant treatment (N = 344 pairs; survivors: 94%, controls: 93%), but more frequently undergo mastectomy (survivors: 81%, controls: 60%) and are less likely to be treated with anthracyclines (survivors: 47%, controls: 66%) or radiotherapy (survivors: 18%, controls: 61%). Despite this, survivors have nearly 3.5-fold (95% CI = 2.17-5.57) greater mortality risk. Here, we show survivors with subsequent breast cancer face excess mortality despite therapeutic tradeoffs and require specialized treatment guidelines.

Presently, >85% of children with cancer will become five-year survivors[1]. As a result of their therapeutic exposures, they are at remarkable risk for long-term health consequences[2,3], with subsequent malignant neoplasms (SMNs) among the most concerning of these. Excluding non-melanoma skin cancer, breast cancer is the most common SMN among adult survivors of childhood cancer[4]. Subsequent breast cancers have been associated with radiation therapy (RT)[5-8], as well as anthracycline chemotherapy exposure[7-10]. The cumulative risk for breast cancer at age 50 among females exposed to chest RT during childhood cancer therapy (30%) is comparable to carrying a *BRCA1/2* gene mutation (up to 31%)[6].

[1]Department of Pediatrics, University of Minnesota, Minneapolis, MN 55455, USA. [2]Department of Hematology/Oncology, Mayo Clinic Arizona, Phoenix, AZ 85054, USA. [3]Department of Medicine, University of Kansas, Westwood, KS 66205, USA. [4]Department of Hematology/Oncology, Northwestern Medicine Lake Forest Hospital, Lurie Cancer Center Affiliate Network, Lake Forest, IL 60045, USA. [5]Department of Medicine, University of Texas Southwestern Medical Center, Dallas, TX 75390, USA. [6]Department of Epidemiology and Cancer Control, St. Jude Children's Research Hospital, Memphis, TN 38105, USA. [7]Department of Pathology and Laboratory Medicine, Children's Hospital Colorado, and Department of Pathology, University of Colorado, Anschutz Medical Campus, Aurora, CO 80045, USA. [8]Department of Pathology & Laboratory Medicine, Nationwide Children's Hospital, Columbus, OH 43205, USA. [9]Department of Epidemiology and Biostatistics, Memorial Sloan Kettering Cancer Center, New York, NY 10017, USA. [10]Department of Pediatrics, University of Chicago, Chicago, IL 60637, USA. [11]Department of Medicine, University of Chicago, Chicago, IL 60637, USA. [12]Department of Medicine, Duke University, Durham, NC 27705, USA. [13]Department of Medicine, University of Minnesota, Minneapolis, MN 55455, USA. ✉e-mail: turc0023@umn.edu

To our knowledge, detailed characterization of treatment decisions for subsequent breast cancer and related toxicities are limited, and treatment guidelines for this population do not exist. Unlike first primary breast cancer, subsequent breast cancer treatment may be complicated by childhood cumulative dose exposure to treatments commonly used to treat breast cancer, including anthracycline chemotherapy and RT. Survivors of childhood cancer are also more likely to have other chronic health conditions[2], which may limit subsequent breast cancer treatments that can be delivered safely and also increase their susceptibility to treatment-related toxicities. For example, cardiovascular late effects including cardiomyopathy and heart failure[11–14] are associated with increasing cumulative exposure to anthracyclines, agents that are also frequently used to treat breast cancer[15,16], but are presumably avoided or limited when treating subsequent breast cancer. While previous studies of subsequent breast cancers have largely focused on survivors of adolescent and adult Hodgkin lymphoma[10,17–20], with the exception of one study[21], inferior survival has been observed compared to first primary breast cancer[18,22]. Differences may be attributable to alterations in breast cancer treatment or related toxicities, but these factors have not been described.

In the present study, we sought to characterize subsequent breast cancer therapeutic modifications and treatment-related toxicities and assess whether these factors contribute to excess mortality among a large population of childhood cancer survivors.

## Results

### Subsequent breast cancers in survivors of childhood cancer
Female survivors of childhood cancer with subsequent breast cancer were identified in the Childhood Cancer Survivor Study (CCSS), a North American multi-institutional retrospective cohort study including 23,558 survivors and designed to quantify and understand the effects of pediatric cancer and treatment on long-term health[23,24]. Among 11,550 females participating in CCSS, 431 had a pathology-ascertained subsequent breast tumor (in situ or invasive) during adulthood (age ≥18 years) occurring at least five years after primary cancer diagnosis (Supplementary Fig. 1). Subsequent breast cancers were diagnosed from 1981–2016. The most common childhood cancer diagnoses in this subgroup were Hodgkin lymphoma (58%), sarcoma (19%) and leukemia (11%) (Table 1). The median latency between childhood cancer diagnosis and breast cancer was 24 years (interquartile range [IQR] = 19–29), and the median age at breast cancer diagnosis was 40 years (IQR = 35–44). Most survivors (69%) were treated with chest RT and 43% received anthracyclines for childhood cancer. Before breast cancer diagnosis, 11% had experienced major cardiovascular events (e.g., myocardial infarction, heart failure) and 5% had non-breast SMNs, most commonly thyroid carcinoma (Supplementary Table 1). Hodgkin lymphoma survivors had greater exposures to chest RT and the shortest median latency to breast cancer diagnosis (23 years, IQR = 18–29) (Supplementary Table 2). Most subsequent breast tumors were invasive (77%); of these, 83% were early stage (I/II). For survivors with available data, 78% had estrogen-receptor (ER)-positive disease and 26% had human epidermal growth factor receptor 2 (HER2)-positive disease.

With additional treatment and acute toxicity data abstracted from medical records, we assessed prescribed subsequent breast cancer treatment information against chronological period-specific National Comprehensive Cancer Network® treatment guidelines[25] for first primary breast cancer. Because breast cancer treatment guidelines permit multiple therapy options[25], we found most survivors (93%) were prescribed guideline-concordant treatment (Supplementary Table 3). Over 60% underwent bilateral mastectomy, but only 12% received any breast and/or axillary irradiation. Nearly 54% were treated with chemotherapy; among these, most received regimens including cyclophosphamide (67%) or taxanes (66%). Nearly two-thirds (66%) of females with ER-positive disease were treated with hormone-modulating therapy.

Features of survivors' personal clinical history were associated with differences in prescribed subsequent breast cancer treatment. Multivariable models adjusted for disease histology/stage demonstrated childhood chest RT dose (none versus up to 35 Gy: OR = 2.35, 95% CI = 1.20–4.57; >35 Gy: OR = 4.87, 95% CI = 2.26–10.50) was associated with breast/axillary RT avoidance (Supplementary Table 4). Anthracycline omission was associated with higher doses of prior anthracyclines (none versus up to 250 mg/m$^2$: OR = 3.06, 95% CI = 1.20–7.77; >250 mg/m$^2$: OR = 5.29, 95% CI = 2.29–12.20) and previous major cardiovascular events (OR = 2.96, 95% CI = 1.05–8.33). Guideline-discordant treatment was associated with treatment with anthracyclines for childhood cancer (OR = 4.18, 95% CI = 1.48–11.85). No survivors with prior SMNs or major cardiovascular events were treated with lumpectomy and radiotherapy (Supplementary Table 5).

### Mortality after subsequent breast cancer
Of the 164 survivors who died after subsequent breast cancer diagnosis, cause of death information ascertained through National Death Index (NDI) linkage and medical record review was available for 135 survivors (Supplementary Table 6). Over 15% of deaths occurred among survivors with in situ carcinomas, of which most (58%) were attributable to non-breast SMNs, primary cancer recurrence, or cardiovascular diseases. Overall, the 15-year breast cancer-specific mortality probability was 19% (95% CI = 15–24%), identical to the corresponding estimate for deaths due to other causes (19%; 95% CI = 15–25%) (Supplementary Table 7). However, breast cancer-specific mortality at 15 years post-diagnosis was higher among survivors with invasive tumors (23%, 95% CI = 18–29%; in situ: 5%, 95% CI = 2–14%) (Fig. 1). The 15-year other-cause mortality probability was higher among survivors with in situ disease (22%; 95% CI = 14–36%), but was still substantial among those with invasive disease (18%; 95% CI = 14–25%). The 15-year breast cancer-specific mortality probability among survivors treated with lumpectomy and RT was 12% (95% CI = 4–37%), identical to mastectomy without RT (12%; 95% CI = 8–18%).

Multivariable models including survivors with complete clinical information assessed all-cause and cause-specific mortality risk factors (n = 314, 97 deaths; Fig. 2). Most survivors who were omitted from these analyses were excluded due to insufficient breast cancer treatment information (69%; Supplementary Fig. 1). We found being prescribed guideline-discordant breast cancer care (HR = 3.04, 95% CI = 1.62–5.71), major cardiovascular events (HR = 1.94, 95% CI = 1.17–3.20), non-breast SMNs (HR = 2.14, 95% CI = 1.17–3.92), and subsequent breast irradiation (HR = 2.34, 95% CI = 1.38–3.97) were risk factors for all-cause mortality. We observed being prescribed guideline-discordant care had an adjusted 7.17-fold greater risk of breast cancer-related death (95% CI = 3.05-16.86). Non-breast SMNs (HR = 2.98, 95% CI = 1.21–7.33), childhood chest RT dose (per 10 Gy, HR = 1.42, 95% CI = 1.14–1.77), and subsequent breast irradiation (HR = 2.33, 95% CI = 1.09–5.01) were risk factors for other health cause-related death.

### Treatment for first primary versus subsequent breast cancer
For comparison, females with first primary breast cancer diagnosed at three academic medical centers were matched one-to-one to CCSS participants diagnosed with subsequent breast cancer. Matching characteristics included breast cancer diagnosis age, year of diagnosis, disease histology and stage, and when feasible, hormone receptor status, HER2 status, and race and ethnicity. Characteristics of the 344 survivor-matched females who developed first primary breast cancer are shown in Supplementary Table 8. Overall, subsequent breast cancer treatment characteristics in the larger sample with non-control-matched survivors (n = 87) were consistent with control-matched survivors (Supplementary Table 9).

**Table 1 | Characteristics of female survivors of childhood cancer who developed subsequent breast cancers (N = 431)**

| Characteristic | N | % or median (IQR) |
|---|---|---|
| Primary cancer diagnosis | | |
| Hodgkin lymphoma | 251 | 57.9 |
| Leukemia | | |
| Lymphoblastic leukemia | 30 | 7.5 |
| Acute myeloid leukemia | 14 | 3.2 |
| Other leukemia | 3 | 0.7 |
| Sarcoma | | |
| Soft tissue sarcoma | 26 | 6.0 |
| Osteosarcoma | 34 | 7.8 |
| Ewings sarcoma | 23 | 5.3 |
| Other bone tumor | 1 | 0.2 |
| Kidney tumor | 19 | 4.4 |
| Non-Hodgkin lymphoma | 18 | 4.2 |
| Neuroblastoma | 5 | 1.2 |
| Central nervous system tumor | | |
| Astrocytoma | 6 | 1.4 |
| Medulloblastoma | 1 | 0.2 |
| Age at childhood cancer diagnosis (years) | 431 | 15 (13-17) |
| Race | | |
| White | 412 | 95.6 |
| Non-White | 19 | 4.4 |
| Childhood cancer treatment[a] | | |
| Treated with chest radiotherapy | 279 | 68.6 |
| Received anthracyclines | 174 | 42.7 |
| Other late chronic health conditions | | |
| Major cardiovascular event before breast cancer | 49 | 11.3 |
| Major cardiovascular event after breast cancer | 35 | 8.1 |
| Subsequent malignancy before breast cancer | 22 | 5.1 |
| Subsequent malignancy after breast cancer | 36 | 8.3 |
| Age at first breast cancer diagnosis (years) | 431 | 40 (35-44) |
| Years between primary and breast cancer diagnosis | 431 | 24 (19-29) |
| Breast cancer treatment era | | |
| Before 2000 | 115 | 26.5 |
| 2000-2009 | 186 | 42.9 |
| 2010-2016 | 130 | 30.6 |
| Subsequent breast cancer stage[b] | | |
| 0 | 101 | 26.3 |
| I | 120 | 31.3 |
| II | 112 | 29.9 |
| III | 34 | 8.9 |
| IV | 14 | 3.7 |
| Breast cancer histology[b] | | |
| DCIS/LCIS only | 101 | 23.5 |
| Invasive | 326 | 76.5 |
| Other breast tumor characteristics[c] | | |
| Progesterone receptor positive | 215 | 66.9 |
| Estrogen receptor positive | 266 | 77.7 |
| HER2 positive | 64 | 25.8 |
| Vital status | | |

**Table 1 (continued) | Characteristics of female survivors of childhood cancer who developed subsequent breast cancers (N = 431)**

| Characteristic | N | % or median (IQR) |
|---|---|---|
| Deceased | 164 | 37.8 |

*CCSS* Childhood Cancer Survivor Study, *IQR* interquartile range, *PNET* primitive neuroectodermal tumor, *DCIS/LCIS* ductal carcinoma in situ/lobular carcinoma in situ, *HER2* human epidermal growth factor receptor 2.
Missing childhood cancer treatment information: chest radiotherapy (n = 27), anthracyclines (n = 20).
Missing breast cancer staging and histology: stage (n = 50), histology (n = 4).
Missing breast tumor information: metastatic breast cancer (n = 27), progesterone receptor status (n = 112), estrogen receptor status (n = 88), HER2 receptor status (n = 186).

Survivors were as likely as controls to be prescribed guideline-concordant treatment (93% versus 94%, P = 0.58) (Fig. 3, Supplementary Table 9). Reasons for receiving guideline-discordant treatment, including treatment refusal, were similar between groups (Supplementary Fig. 2). Survivors had 5.92-fold greater odds of undergoing bilateral mastectomy (95% CI = 3.87-9.04; 62% versus 24%), and correspondingly were less likely to undergo lumpectomy (30% versus 48%, P < 0.001) and receive breast and/or axillary irradiation (28% versus 61%, P < 0.001). Treatment with chemotherapy (yes/no) did not differ (55% versus 56%, P = 0.43), but survivors were less likely to be treated with cyclophosphamide (67% versus 76%, P = 0.028) and doxorubicin (43% versus 64%, P < 0.001) and more likely to be treated with taxanes (72% versus 63%, P = 0.033) than controls. Among females with ER-positive disease, survivors were less likely to receive hormone modulating therapy (68% versus 78%, P < 0.01).

Among those treated with chemotherapy and prescribed guideline-concordant care for breast cancer, survivors and controls had similar rates of any treatment modification (77% versus 74%, P = 0.22) (Supplementary Table 10), but survivors were more likely to have a drug omission (21% versus 8%, P < 0.01). Survivors were also more likely to experience at least one acute treatment-related toxicity than controls (86% versus 74%, P = 0.029). Specifically, survivors were more likely to develop cytopenias (20% versus 8%, P = 0.017) and neurotoxicity (30% versus 17%, P = 0.038), and had suggestively greater frequency of cardiotoxicity and gastrointestinal toxicity compared to controls.

**Mortality after first primary versus subsequent breast cancer**
In analyses with matched controls with medical record- or registry-ascertained mortality data (n = 241 pairs with 84 deaths among survivors and 33 deaths among controls), the 15-year mortality probability after breast cancer diagnosis was 41% among survivors (95% CI = 32-49%) versus 14% in controls (95% CI = 9-19%; HR = 3.48, 95% CI = 2.19-5.54; Fig. 4, Supplementary Table 11). This excess mortality risk between survivors and the general population was substantial among those with in situ disease (HR = 9.94, 95% CI = 3.13-31.57) and persisted among those with invasive disease (HR = 2.88, 95% CI = 1.77-4.71). Sensitivity analyses to assess potential residual confounding after matching showed comparable results (Supplementary Table 12). Breast cancer treatment modifications and toxicities did not entirely explain survivors' excess mortality risk. Among survivors and controls with invasive tumors, neither of these factors were independently associated with increased all-cause mortality risk in multivariable modeling (Supplementary Table 13). The mortality cumulative incidence among controls was consistent with crude survival probabilities among females with first primary breast cancer in the US Surveillance, Epidemiology, and End Results (SEER) Program database (Supplementary Table 14).

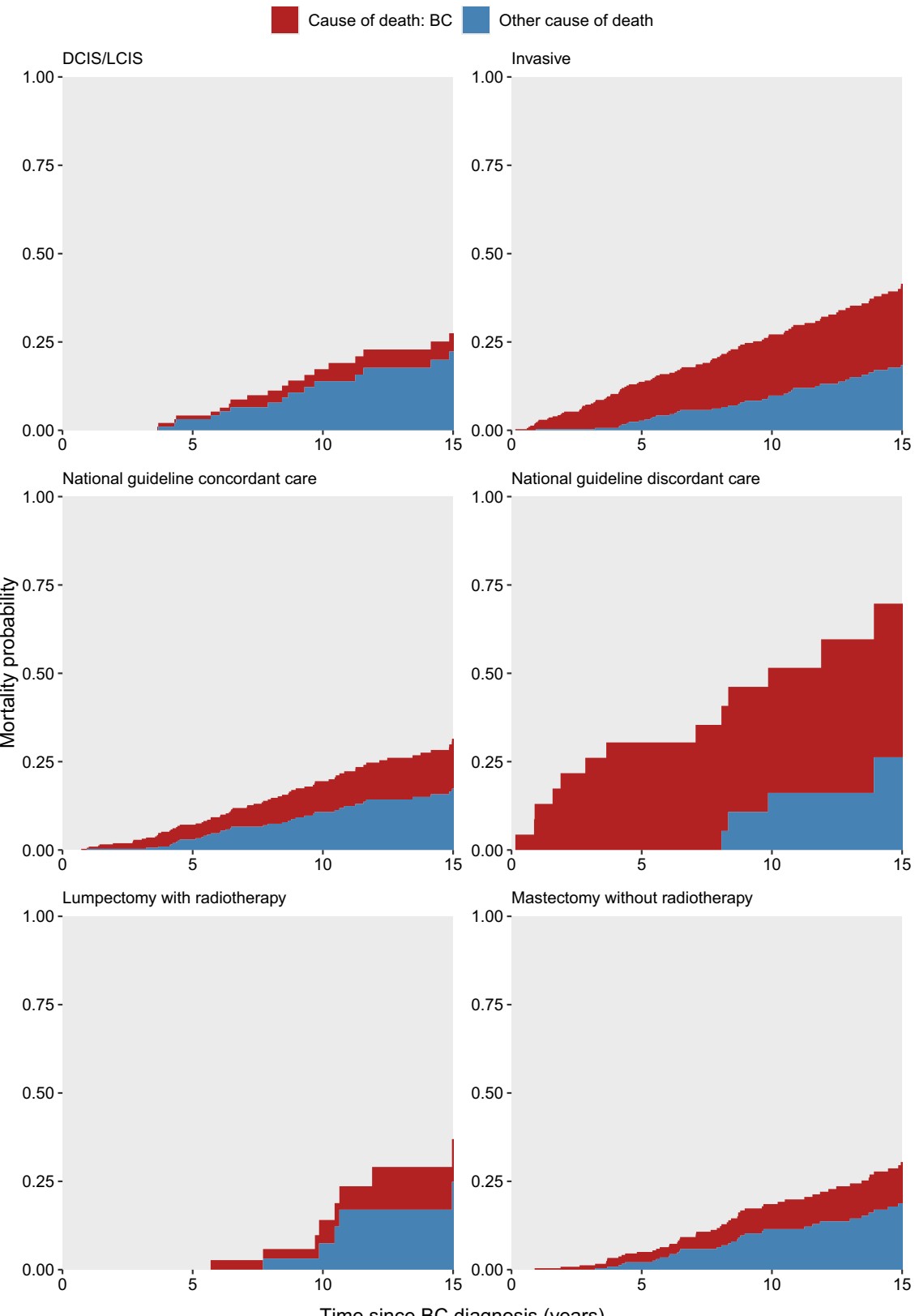

**Fig. 1 | Cumulative mortality partitioned by cause of death after subsequent breast cancer among childhood cancer survivors.** Mortality probabilities due to breast cancer (BC) are shown relative to other causes in red and blue, respectively ($n = 402$ overall). Top row panels are stratified by breast cancer disease histology (DCIS/ LCIS: ductal or lobular carcinoma in situ). Middle row panels are stratified by concordance/discordance with temporally matched national breast cancer treatment guidelines. Bottom row panels are stratified by treatment modality.

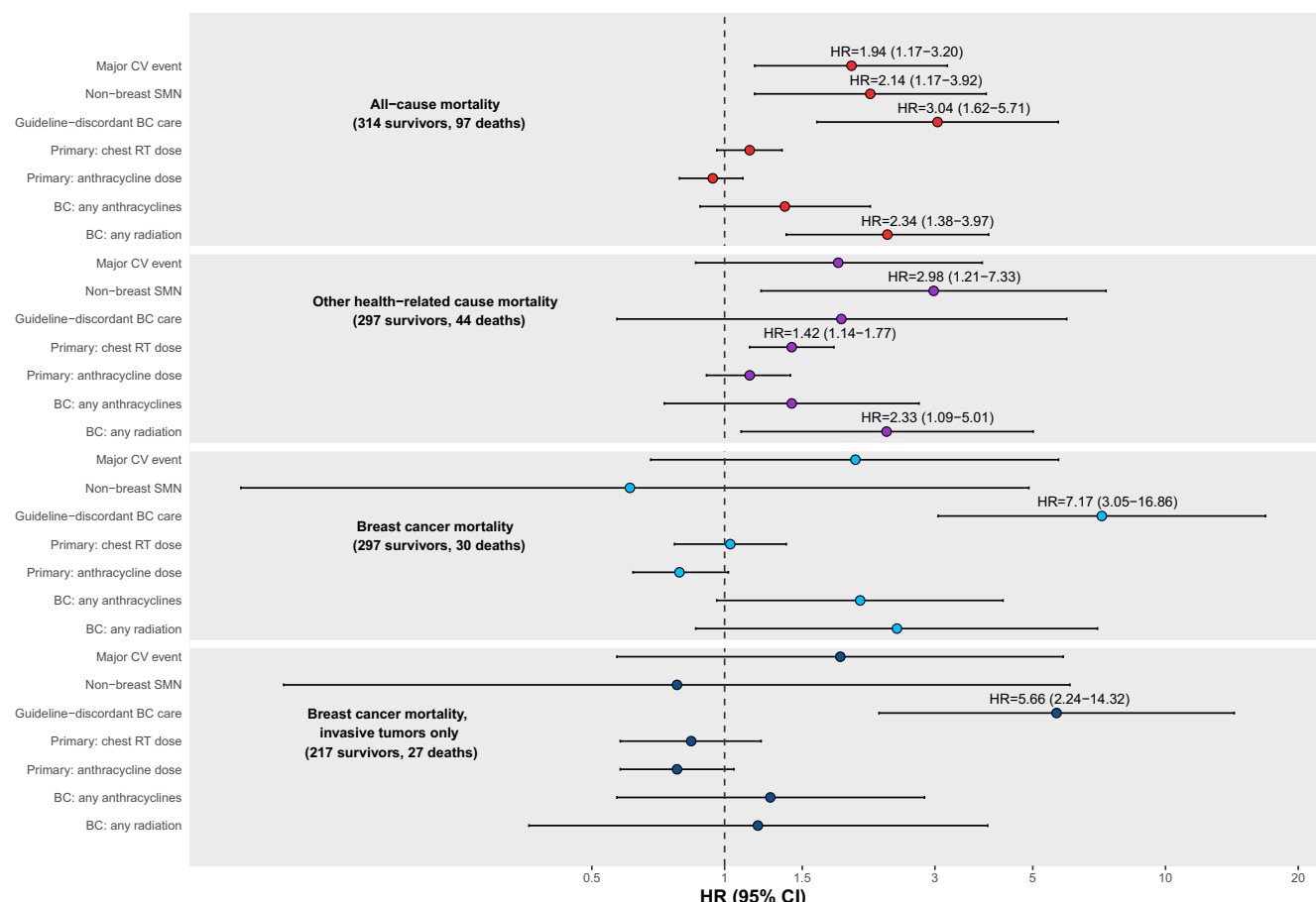

**Fig. 2 | Risk factors for all-cause and cause-specific mortality after subsequent breast cancer in childhood cancer survivors.** BC breast cancer, RT radiation therapy, CV cardiovascular, SMN subsequent malignant neoplasm. HRs and corresponding 95% confidence intervals (CIs, shown as error bars) from multivariable Cox proportional hazards regression models evaluating associations between primary and breast cancer treatments and the mortality hazard rate using age as the time scale and adjusting for covariates included above are shown, as well as breast cancer diagnosis year and histology (invasive versus non-invasive) or stage (III/IV versus I/II, invasive carcinomas only) (*n* = 314 overall). HRs and 95% CIs are annotated for statistically significant variables (two-sided *P* < 0.05). Primary cancer chest RT dose is provided per 10 Gray and primary anthracycline dose is provided per 100 mg/m².

## Discussion

This study represents a comprehensive characterization of subsequent breast cancer treatment, toxicity, and mortality in a large cohort of childhood cancer survivors. With the exception of a previous Hodgkin lymphoma survivor case series with breast cancers diagnosed before 1997[19], prior studies of subsequent breast cancer treatment have lacked information about specific anticancer agents[17,18,21,22]. Furthermore, subsequent cancer-specific treatment and outcome data are not systematically captured in established childhood cancer survivor cohorts like CCSS. Leveraging detailed information about chemotherapy, radiation, hormone-modulating therapy, and HER2-targeted agents, we found that most childhood cancer survivors were prescribed guideline-concordant breast cancer treatment with similar frequency to controls, albeit with apparent therapeutic tradeoffs considering childhood cancer treatment and comorbidity history. Despite these therapeutic tradeoffs, survivors had a 3.5-fold greater risk of death overall following breast cancer compared to matched controls, with substantial mortality burden due to other causes, particularly among females with in situ disease. This exceeds the previous estimate (HR = 2.4; 95% CI = 1.7–3.2)[22], and considers treatment of breast cancer consistent with national guidelines, longer follow-up, and more contemporary subsequent breast cancer diagnoses. We also found survivors were more likely to have drug omissions and acute toxicities than controls. While these may be mortality risk factors, they were not associated with all-cause mortality risk in these data.

This study characterizes use of anthracycline-containing chemotherapy regimens for subsequent breast cancer. Anthracyclines are commonly used to treat childhood cancer but are associated with serious long-term cardiovascular toxicity[11–14]. Cumulative anthracycline exposure ≥250 mg/m² among survivors is associated with a ninefold increased risk for cardiomyopathy[26], and a lifetime cumulative dose limit of 400-550 mg/m² is recommended[27]. However, anthracyclines are effective and frequently used drugs for breast cancer[15,16], especially early-stage disease[28]. The implications of avoiding or administering anthracyclines for subsequent breast cancers remains unclear. While associations between anthracyclines administered for childhood cancer or subsequent breast cancer and increased risk of death due to other causes were not observed among survivors, we note that the lack of cumulative anthracycline dose information across SMN episodes and self-report of cardiovascular late effects in CCSS limit these conclusions. However, based on the SMN types observed prior to breast cancer, it is not anticipated that most would have yielded additional anthracycline exposure. Anthracycline use decision-making may be improved with precision cardiovascular late effects risk stratification[29–31]. Dexrazoxane has also been used for cardioprotection with doxorubicin-based combination therapy for advanced breast cancer[32,33], although evidence supporting the use of cardioprotective agents is limited in survivors[34].

The increased risks for SMNs among childhood cancer survivors are well documented[2,4,35,36], as is the increased risk for mortality

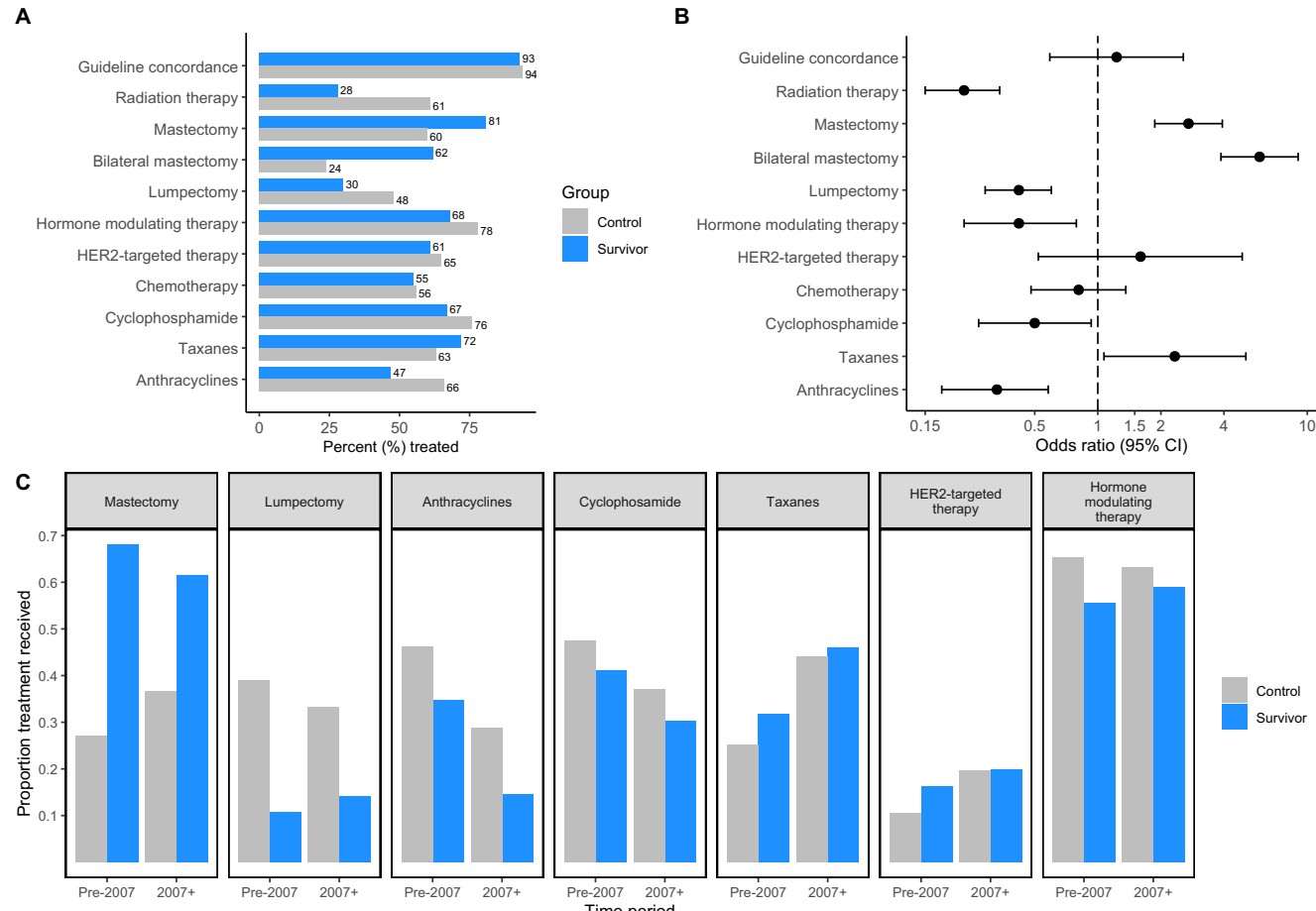

**Fig. 3 | Differences in the management of breast cancers in childhood cancer survivors and the general population (controls).** For all panels, up to 344 survivor-control pairs were assessed. Percentages treated with specific modalities are provided in panel A for survivors (blue) and matched controls (grey); percentages for mastectomies and lumpectomies are among those who received surgery for breast cancer, and percentages treated with cyclophosphamide, taxanes, anthracyclines are among those treated with chemotherapy for breast cancer. Mastectomy refers to unilateral or bilateral mastectomy unless otherwise specified. Guideline concordance refers to standard of care for breast cancer consistent with temporal national guidelines. In panel B, the odds of treatment receipt in survivors are compared with matched controls (odds ratios or ORs; point estimates to the right of the dashed line reflect increasing survivor odds of treatment) and corresponding 95% confidence intervals are shown (CIs; shown as error bars). Panel C shows temporal changes in breast cancer treatment decisions for survivors and matched controls; 2007 was the median year of breast cancer diagnosis, each bar reflects the percent treated among all participants with non-missing values for a given treatment within the time window specified in the x-axis.

following SMNs[37,38]. While Moskowitz et al.[22] also reported a high risk for other-cause death among survivors with subsequent breast cancer, non-breast SMNs were not assessed separately from other chronic health conditions. We found that non-breast SMNs were significantly associated with increased risks for mortality overall and other-cause mortality. Among survivors with a documented non-breast SMN cause of death, the SMNs occurred after breast cancer and included soft tissue sarcoma, acute myeloid leukemia, lung cancer, and colorectal cancer. It remains unclear whether the selection of treatments for subsequent breast cancer considered other SMN risks. Cyclophosphamide to treat subsequent breast cancer was less common among survivors, which may in part be due to reports of alkylating agents as risk factors for SMNs[39–41]. Further study of how the treatment of subsequent breast cancers may be balanced considering risks of developing other SMNs is needed.

Survivors overwhelmingly received mastectomy without breast irradiation compared to the general population. Prior RT is considered a contraindication to breast-conserving surgery with radiation[19,42,43], making mastectomy the standard surgical recommendation among survivors. RT carries risks for compromised wound healing[44] and has been associated with adverse breast reconstruction outcomes[19,45].

However, studies in Hodgkin lymphoma survivors suggest breast-conserving therapy including repeat irradiation may be feasible without apparent short-term adverse consequences[46,47]. Given that our data support previous observations that neither mastectomy without RT nor lumpectomy with RT demonstrate superior cause-specific survival[47], less aggressive surgical management may be considered, especially among survivors with in situ disease or who received lower doses or volumes of chest-directed RT.

Study limitations must be acknowledged. Because these data reflect the experience of five-year childhood cancer survivors with subsequent breast cancer and females with first primary breast cancer treated in the US and Canada, the generalizability of our findings to other countries is limited. Data were collected retrospectively and may not fully capture treatment decision-making considerations. For example, the Oncotype DX score, which is considered an important prognostic tool for treatment decisions, was not available for cases or controls. Including female survivors with in situ subsequent breast cancers affect estimates of excess mortality. These were included given previous evidence indicating their increased risk of mortality especially for non-breast cancer-related causes[22]. We conducted analyses limited to and excluding those with in situ carcinomas; notably,

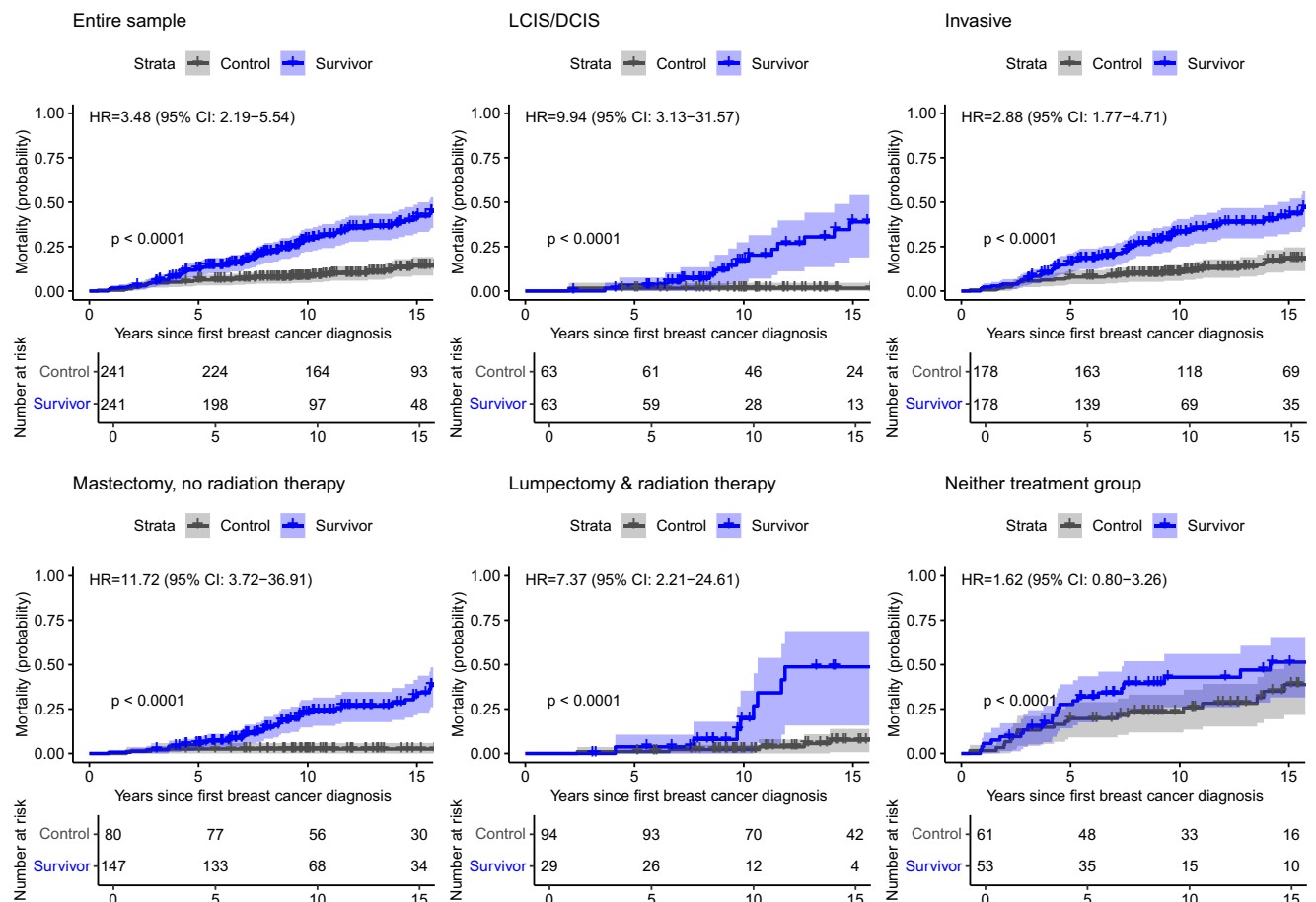

**Fig. 4 | All-cause mortality probabilities in childhood cancer survivors and matched controls by breast cancer histology and treatment modality.** In all panels, mortality curves for survivors are shown in blue while those for matched controls are shown in gray, and hazard ratios (HRs) adjusted for guideline-concordant breast cancer treatment comparing mortality risks for survivors and controls are provided along with corresponding 95% confidence intervals (CIs) (n = 241 survivor-control pairs overall). Differences in mortality curves for survivors and matched controls were evaluated with two-sided log-rank tests with robust variance estimation; these p values are shown in the lower left quadrant for each panel.

we observed conditional excess mortality persists after excluding in situ carcinomas. It is possible that cause of death may be misclassified for some subsequent breast cancer patients[48] and there were differences in methods to ascertain deaths among survivors and controls. Cause of death for survivors was documented in multiple sources, including NDI, increasing likelihood of accuracy. We also observed the crude survival estimates in SEER and matched controls in these data are largely consistent, especially for in situ breast cancer. The procedure to match females with first primary breast cancer to CCSS survivors was imperfect, leading to the inclusion of fewer White matched controls. Sensitivity analyses where we also adjusted for matching variables led to similar results. These data are also limited by missingness. For example, in multivariable analyses of mortality risk factors among survivors with subsequent breast cancer (Fig. 2), 117 survivors were excluded due to data missingness. Nearly half of those excluded were treated for breast cancer before 2000; accordingly, the most common reason for exclusion was related to incomplete breast cancer treatment documentation. However, the survivors included in this mortality risk factor analysis had high overlap with and are representative of the survivors assessed in mortality analyses with matched controls, whereas excluded survivors had higher mortality (54%, 95% CI = 43–63%). Because Black females with breast cancer have consistently worse survival[49,50] and also constituted the largest subgroup of non-White matched controls and excluded survivors had a higher rate of death, the excess mortality reported here may be underestimated. This bias may be tempered by the use of controls drawn from US academic medical centers, which typically have lower cancer mortality rates than community-based hospitals[51,52].

Genetic susceptibility for breast cancer and its impact on survival was not assessed. Despite the potential for inferior outcomes among younger females with first primary breast cancer[53–55] or with genetic breast cancer predisposition[56], these results reflect analyses where survivors and controls were matched based on diagnosis age, disease stage, and if available, molecular subtype. Our analyses also did not account for all hypothesized prognostic predictors, e.g., novel biomarkers, lifestyle and reproductive factors, given their limited availability across cases and controls, and instead limited our analyses to consider classical clinical and pathological features predictive of survival, including histology, stage, and molecular subtype. Another important limitation of the present study is the lack of cause of death information among matched controls. We found breast cancer was the predominant cause of death among survivors with invasive tumors, including survivors diagnosed with early-stage (I/II) disease. While we would expect excess all-cause mortality to persist when comparing survivors and matched controls with invasive breast cancer, given the greater contribution of non-breast cancer-related deaths among survivors, we observed considerably higher all-cause mortality probabilities at time points closer to breast cancer diagnosis among survivors with invasive versus in situ carcinomas (5-year: 16% versus 3%; 10-year: 33% versus 17%). We speculate that these findings may reflect excess subsequent breast cancer-related deaths among childhood cancer survivors despite therapeutic tradeoffs. This is consistent with

previous work that reported a modest increase in breast cancer-related death among survivors, albeit without consideration of detailed breast cancer treatment data[22]. Further study is needed to assess the magnitude of excess subsequent breast cancer-related death considering treatment modifications and acute toxicities, but requires cause of death and suitable breast cancer treatment data in a comparison group of females with first primary breast cancer.

Here, we have provided a large and comprehensive description of subsequent breast cancer treatment, toxicity, and mortality described to date among survivors of childhood cancer. Although survivors are largely prescribed guideline-concordant treatment for breast cancer, therapeutic tradeoffs do not appear to mitigate survivors' excess mortality burden, nor do more frequent drug omissions and treatment-associated toxicities adequately explain the excess mortality risk. These observations underscore the complexity of balancing decisions to treat current disease while effectively managing survivors' comorbidities, and the need for specialized treatment guidelines for subsequent breast cancer.

## Methods

### Childhood Cancer Survivor Study (CCSS) participants

The CCSS is a North American multi-institutional retrospective cohort study with longitudinal follow-up of five-year survivors of childhood cancer. CCSS participants were diagnosed at age <21 years between 1970–1999. Details of the CCSS methodology are published elsewhere[23,24]. Human subjects research approval was granted prior to recruitment by the Institutional Review Boards of all 31 participating CCSS institutions and the institutions where controls were recruited: the University of Minnesota, Duke University, and the University of Chicago. All participants provided written/online/verbal informed consent. Participants received no compensation for their inclusion in this study.

For CCSS participants, childhood cancer diagnosis and treatment data, including cumulative chest RT and anthracycline dose, were abstracted from medical records using standardized protocols[23,24]. Cumulative anthracycline doses within five years of primary diagnosis were quantified as doxorubicin-equivalent doses[57]. Individual radiation therapy (RT) records were centrally reviewed and the cumulative chest RT dose was estimated as the total delivered dose within 5 years of primary diagnosis from all overlapping RT fields[58].

Information about chronic health conditions were obtained from CCSS baseline and follow-up questionnaires (accessible at http://ccss.stjude.org). Major cardiovascular events included self-reported heart failure (HF), coronary artery disease (CAD) resulting in myocardial infarction or coronary intervention, or stroke that were centrally graded based on the modified National Cancer Institute Common Terminology Criteria for Adverse Events (CTCAE, version 4.03). These conditions were specifically defined as HF with grades ≥ 3 (requiring medications or heart transplant, or resulting in death), CAD with grades ≥ 3 (requiring anti-anginal medication use or intervention, e.g., catheterization, angioplasty, bypass graft, or resulting in death), and stroke with grades ≥ 4 (life-threatening or fatal), occurring ≥ 5 years post-childhood cancer diagnosis. SMNs were identified by self-/proxy-report or death certificates and validated via pathology report, medical record, and/or death certificate review. Neoplasms classified with a behavior code of 3 by the International Classification of Diseases for Oncology[59] (third edition) that were histologically unique from childhood cancer were considered SMNs.

Sex was considered in the design of this study and was based self-/proxy-report or information reported from the treating institution. Female survivors with a confirmed breast cancer diagnosis (ductal carcinoma in situ [DCIS], lobular carcinoma in situ [LCIS], or invasive carcinoma) as of December 31, 2016 at age ≥18 years and ≥5 years after initial childhood cancer diagnosis were included. Six participants with ambiguous diagnoses were excluded (3 sarcoma, 1 benign

fibroadenoma, 2 phyllodes tumors), leaving 431 females with subsequent breast cancer (in situ or invasive) for inclusion in this analysis.

Breast cancer diagnosis, treatment, and toxicity data were collected from medical records using a standardized abstraction form. Breast cancer histology, stage, size, location, hormone receptor status, and human epidermal growth factor receptor 2 (HER2) status were abstracted, along with treatments for the first breast cancer, including surgery type, cumulative radiation dose and site, cumulative chemotherapy doses, hormone modulating therapy, and HER2-targeted therapy. Prescribed breast cancer treatment information was reviewed against published national breast cancer treatment guidelines (National Comprehensive Cancer Network®)[25] from the year of treatment for cancers diagnosed from the time the guidelines were established in 1995 through 2016. Consistency with guideline-concordant breast cancer treatment was defined by matches in treatments prescribed and guideline-listed treatments for the relevant time period, with consideration of treatments based on disease stage, hormone receptor and HER2 status, when applicable. If dose modifications were required due to toxicity, but the initial prescribed dose was appropriate, therapy was considered guideline-concordant. If individuals were documented as participating in a clinical trial, treatment was considered guideline-concordant. In situations of uncertainty regarding guideline concordance, or for cancers diagnosed before 1995, relevant contemporary breast cancer treatment literature was reviewed. If the standard of care could not be clearly established or treatment data was incomplete or unknown, guideline concordance was considered unknown.

If therapy was refused or if treatment was deemed unsuitable/unsafe by the physician, this was recorded. Treatment modifications, including drug/cycle omission, treatment delay, and granulocyte colony-stimulating factor receipt, were evaluated among those treated with chemotherapy. Acute treatment-related toxicities, including surgical complications, fever/neutropenia hospitalizations, infections, cytopenias, and neurological, cardiac, pulmonary, gastrointestinal, dermatological, and musculoskeletal toxicities were assessed. To minimize bias, 10% of records were randomly sampled and reviewed by two independent oncologists (LMT; JPN). Differences in abstractions between reviewers were reconciled.

Deaths in CCSS were ascertained by medical record and National Death Index (NDI) review (completed December 31, 2017). Causes of death were adjudicated with standardized rules based on the International Classification of Diseases (ninth and tenth revisions).

### Control participants

A multi-institutional comparison group of women with first primary breast cancer diagnosed at the University of Minnesota ($n = 120$), Duke University ($n = 112$), and the University of Chicago ($n = 112$) health systems was assembled. A total of 344 females diagnosed with primary LCIS/DCIS ($n = 85$) or invasive ($n = 259$) breast cancer were matched one-to-one to CCSS participants by breast cancer diagnosis age, diagnosis year, and disease histology and/or stage. Because widespread testing for estrogen receptor status only began in the mid-1990s, hormone receptor and HER2 status were not consistently available for participants diagnosed before 1995. Therefore, when such data were available, controls were also matched on hormone receptor status and HER2 status. Race and ethnicity was also matched when multiple controls could be matched on breast cancer disease characteristics. When multiple suitable matches considering all matching factors were available, the diagnosis date closest to that of the survivor case was selected. Medical record abstraction followed the same procedures implemented for CCSS participants. Deaths were ascertained from medical record review, institutional databases, and cancer registry data. Causes of death were unavailable for matched controls. Matched survivor-control pairs where any member of the pair with unknown vital status information were excluded from mortality analyses.

## Statistical analysis

Analyses restricted to CCSS survivors were conducted to evaluate factors associated with prescribed subsequent breast cancer treatments and mortality risk. Logistic regression models evaluated associations between prescribed breast cancer treatments (i.e., care discordant with national care guidelines; radiotherapy avoidance; anthracycline avoidance) and childhood cancer treatments (chest radiotherapy dose, categorized as none, >0 to ≤35 Gy, and >35 Gy; anthracycline dose, categorized as none, >0 to ≤250 mg/m², and >250 mg/m²) among survivors, adjusting for breast cancer diagnosis age and year, previous history of major cardiovascular events and non-breast SMNs, and breast cancer histology (or stage, for analysis of survivors with invasive breast cancer only). For mortality analyses, the index event was breast cancer diagnosis with time at risk ending at death or censoring for participants who were alive at the time of data abstraction (follow-up ending on December 28, 2022). Breast cancer-specific mortality cumulative incidence was estimated using a non-parametric estimator, accounting for competing risks posed by other causes of death[60]. Cox proportional hazards models for all-cause and cause-specific mortality using age as the time scale[61] were used to estimate hazard ratios (HRs) for guideline-discordant breast cancer treatment and childhood cancer treatments (continuous chest radiotherapy and anthracycline doses with one degree-of-freedom terms), adjusting for breast cancer histology and diagnosis year, breast cancer treatments (any radiotherapy; any anthracyclines), and time-varying non-breast SMNs and major cardiovascular late effects. For cause-specific mortality HRs, survivors without cause of death information were excluded and deaths that were not cause-specific led to the removal of affected subjects from the at-risk pool.

We also performed analyses comparing survivors with subsequent breast cancer and matched controls with first primary breast cancer. Univariate conditional logistic regression models evaluated differences in prescribed breast cancer treatment modalities conditioned on matched pairs. Odds ratios for breast cancer treatment modifications and acute toxicities comparing survivors with matched controls were obtained, specifically among those treated with chemotherapy and prescribed guideline-concordant treatment, using logistic regression models adjusted for breast cancer diagnosis age, year, histology and race/ethnicity. Cumulative all-cause mortality probabilities were estimated with the Kaplan-Meier method and log-rank tests with robust variant estimation[62] compared cumulative incidence curves. Cox regression models with robust variance estimation[62] were used to compare mortality hazard rates between survivors and controls, adjusting for receipt of guideline-concordant breast cancer treatment. A sensitivity analysis to assess potential residual confounding after matching was conducted, i.e., evaluating mortality hazard ratios also adjusted for breast cancer diagnosis age and diagnosis year, histology, and race/ethnicity, along with receipt of guideline-concordant breast cancer treatment. This multivariable model was then further adjusted for experiencing any treatment modification or acute toxicity to assess whether these factors attenuated differences in mortality risk between survivors and controls. For comparison, we evaluated crude mortality from the Surveillance, Epidemiology, and End Results (SEER) 8 Program database (1975–2020), including 98,488 females whose first cancer was an in situ or invasive breast cancer diagnosed between 1981 and 2016 at ages 35 to 49 years.

All analyses are complete case analyses and were performed using R v.4.2.2. All *p* values presented were based on two-sided tests unless otherwise specified. All descriptive statistics and model results accounted for sampling weights reflecting the undersampling of acute lymphoblastic leukemia survivors (by CCSS study design).

## Reporting summary

Further information on research design is available in the Nature Portfolio Reporting Summary linked to this article.

## Data availability

The Childhood Cancer Survivor Study is a US National Cancer Institute funded resource (U24 CA55727) to promote and facilitate research among long-term survivors of cancer diagnosed during childhood and adolescence. CCSS data are publicly available on dbGaP at https://www.ncbi.nlm.nih.gov/gap/ (accession number phs001327.v2.p1) and on the St. Jude Survivorship Portal within the St. Jude Cloud at https://survivorship.stjude.cloud/. Those interested in using this resource are encouraged to visit http://ccss.stjude. The deidentified, processed data used in this study have been deposited in a public repository (https://doi.org/10.5281/zenodo.14989624). The raw data are protected and are not available due to data privacy laws. All remaining data supporting the findings of this study can be found in the Article, Supplementary Material files and Source Data file. Source data are provided with this paper.

## Code availability

All code packages are publicly available and include: survival (https://cran.r-project.org/web/packages/survival/index.html), survminer (https://cran.r-project.org/web/packages/survminer/index.html), and ggplot2 (https://cran.r-project.org/web/packages/ggplot2/index.html).

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

## Acknowledgements

This work was funded by the US National Cancer Institute (K08 CA234232, LM Turcotte, principal investigator; U24 CA55727, GT Armstrong, principal investigator) and the American Lebanese Syrian Associated Charities. Support to C Im was provided by NCI R21 CA261833. Support to CS Moskowitz was provided by NCI P30 CA008748.

## Author contributions

LMT, JPN, YY, and LLR designed the study and developed the concept. CI and LMT provided supervision for the study. CI, ES, SM, RR, JS, HW, AJM, VN, MAA, MRC, GTA, RN, KCO, JPN, AB, and LMT collected the data. CI, HH, ZL, and LMT prepared and analyzed the data. YY and CSM provided insights regarding statistical interpretation. CI and LMT drafted the manuscript. Funding for this project was acquired by LMT, LLR, and GTA. All authors including LGS and TOM contributed critical revisions and approved the final manuscript for publication.

## Competing interests

The authors declare no competing interests.
