## [Transparent Peer Review file · Nature Communications]

Treatment, toxicity, and mortality after subsequent breast cancer in female survivors of childhood cancer

Corresponding Author: Dr Lucie Turcotte

Version 0:

Reviewer comments:

Reviewer #1

(Remarks to the Author)

In their study, Im et al describe therapeutics, treatment-related toxicities and mortality in 431 female childhood cancer survivors from the Childhood Cancer Survivor Study (CCSS) who later developed breast cancer ("survivors") and compare treatment and survival characteristics in these patients to matched females with primary adult breast cancer ("controls") and to population-based SEER rates. They found that survivors are often treated differently for their breast cancer than controls and that survivors have increased mortality compared to controls, both overall and stratified by breast cancer characteristics. The results are interesting and have the potential to advance the science, but the manuscript suffers from lack of focus and flow and from data missingness, making interpretation of results very challenging.

Major comments

- The manuscript is extremely dense with many analyses being carried out, making it challenging to follow. Moreover, the results section occasionally moves back and forth between tables and figures in a somewhat random pattern. For instance, in the "Treatment Selection, modification and toxicity section", the authors refer to Supplementary Table 4, then Supplementary Table 5, then Supplementary Figure 1 and then back again to Supplementary Table 4 when comparing survivors to matched controls. Following this, the last sentence in the same paragraph refers to Supplementary Table 6, which does not include the matched controls. This back and forth severely compromises the readability. The manuscript would benefit from a more focused and linear approach.
- Keeping with the theme of the comment above: Supplementary Tables 1 and 2 are among the survivors only, Supplemental Tables 3-5 include both survivors and matched controls, Supplemental Tables 6-8 are among the survivors only, Supplemental Tables 9-12 include both survivors and matched controls, Supplemental Tables 13-14 are among the survivors only, Supplemental Tables 15 and 16 include both survivors and matched controls, Supplementary Table 17 compares survivors to SEER data, and finally Supplementary Table 18 includes both survivors and matched controls. This whiplash approach makes the manuscript far too complicated to follow.
- More detailed information on the matching approach should be provided in the supplementary material. I'm unsure what they authors mean by "ordered by priority." Does this mean that not all matching variables were used for each match?
- The authors state in the statistical methods that conditional logistic regression analysis was used to compare survivors to matched primary cancer controls, sometimes adjusting for other characteristics and sometimes not. I could not determine in what specific adjustments were made in each of the numerous tables and figures comparing survivors to their matched controls.
- Table 2 ("Risk factors for all-cause, health-related cause, and breast cancer-specific mortality in childhood cancer survivors") reports 99 deaths overall, but only 43 for non-breast cancer related mortality and 33 for breast cancer related mortality. This only sums to 76 deaths. Please verify that these numbers are correct and presumed missingness was based in inability to ascertain cause of death. If this is indeed true, how did the authors account for missing cause of death when evaluating associations with cause-specific mortality?
- Table 2 includes continuous effects (breast cancer diagnosis year, chest RT dose, anthracycline dose): I assume these associations were fit using a one degree-of-freedom term? If so, this should be noted in the statistical methods. Modeling continuous effects in this manner assumes a log-linear dose-response association between exposure and risk of death. Did the authors examine the functional form of association to ensure that modeling assumptions were met? Also, how did the authors model subjects who did not receive anthracycline or chest RT? Were they given a value of 0? I could imagine a scenario where risk or death increases dramatically in patients not receiving any anthracycline or chest RT to those

receiving one dose, with more modest increases as dosage increases, which would be counter to the log-linear association modeling assumption.

- The authors report that 164 of the 431 CCSS subjects subsequently died, but Table 2 suggests that only 320 subjects were used in the mortality analyses, among whom 99 died. My guess is that the discrepancy is because of dropout due to missing values for some of the variables of interest, but the authors need to provide more details. Moreover, the death rate differs among those included and not included in Table 2: of the 320 included in Table 2, 99 subsequently died (31%), whereas among those not included in Table 2, 59% $((164-99)/(431-320))$ subsequently died. This raises serious concerns about the representativeness of the subjects used in Table 2 analyses.

- This missingness issue extends to other tables and figures as well. The authors note this as a limitation in the discussion section, but I'm concerned it may have a profound effect on the results that has the potential to alter interpretation of results.

- The manuscript would greatly benefit from a CONSORT diagram that displays the beginning sample size, exclusions, and number included in each of the analyses presented in the tables and figures.

Minor comments

- Supplementary Table 3. Even though the authors attempted to match on race/ethnicity, there seem to be large differences in race between the survivors and the matched general population subjects. The authors should acknowledge this in the text of their manuscript.

- The authors should justify why they are including LCIS cases in addition to DCIS and invasive breast cancer.

Reviewer #2

(Remarks to the Author)

The comparison group for the CCSS was a multi-institutional group of females with primary breast cancer diagnosed at three academic medical centers. Do the authors believe that the outcomes of breast cancer at top academic medical centers in the US are slightly better than the overall national average? If yes, then the difference between the control and experimental group would be higher than expected. Please give national stats for the breast cancer outcomes. The 1:1 match is well noted.

The external validity of the study is poor, since the long term survivors do not fare as well as in the US (compared to developing countries). Please add a section on this in point in the discussion

Some of the risk factors are not described. In fact, <25% of the overall breast cancers may be attributed to inheritance in some populations (e.g. BRCA), and the genome is mentioned in the manuscript. However, majority of the breast cancers are due to epigenetic changes. We have abundant published data that certain common risk factors increase the risk of breast Ca - alcohol, fast food, obesity, (smoking association is weaker), early menarche, late menopause, absence of breast feeding, having kids at a very late age etc. We also have abundant data on BMT and cancer survivors showing multiple psychologic comorbidities. Thus the most important question is that whether the risk of increased breast cancer is due to cancer treatments, or actually due to behavioural changes due to cancer in these survivors, or both? Please explain how can we exclude confounders and effect modifiers in the absence of data present on epigenetic risk factors.

Reviewer #3

(Remarks to the Author)

Thank you for inviting me to review the manuscript titled "Treatment, Toxicity, and Mortality After Primary Versus Subsequent Breast Cancer: A Report from the Childhood Cancer Survivor Study." The authors have explored a significant research question. I have several major and minor comments that should be addressed to enhance the manuscript's clarity and quality.

Abstract, Page 3

The abstract needs some rewording to be more informative and clearer.

Background, line 64: If the wording allows, please add 'particularly those who received chest radiation' after 'Childhood cancer survivors.' So, the sentence would read as follows: "Childhood cancer survivors, particularly those who received chest radiation, are at high risk for developing subsequent breast cancer with higher mortality than females with first primary breast cancer."

Methods, line 67: Please mention the number of female five-year survivors of childhood cancer identified from the multicentre retrospective cohort study, as well as the number of controls (females with first primary breast cancer).

Line 69: For clarity, please add 'first' before 'primary.' So, the sentence would read as: "and breast cancer characteristics with females with first primary breast cancer..."

Line 72: As mentioned above, add 'first' before 'primary breast cancer.' Please correct this throughout the manuscript, so it's clear to the readers that it was the first cancer diagnosed in controls.

Line 79: How many deaths were among controls?

Introduction, Page 4

Lines 101-102: 'Survivors of childhood cancer are also more likely to have other chronic health conditions'. Please provide examples of chronic conditions in survivors that can affect treatment.

Patients and Methods, Page 5:

Lines 118-119: Please add a short paragraph about Childhood Cancer Survivor Study. Saying that 'details of the CCSS methodology are published elsewhere' without providing any information about the aim of the study and number of survivors recruited is not acceptable.

Lines 126-131: The authors mentioned that neoplasms classified with a behaviour code 3 (i.e., invasive cancers) were considered subsequent malignant neoplasms. They also included in situ diseases. However, it is unclear why non-invasive breast cancers (such as DCIS and LCIS) were included. Excluding in situ cancers from survival analysis is generally recommended because they are non-invasive and typically have different prognoses and treatment approaches compared to invasive cancers. Including them might skew survival statistics.

Statistical analysis

Lines 162-163: For mortality analyses, the index event was breast cancer diagnosis with time at risk ending at death or censoring for participants who were alive at the time of data abstraction. In the above sentence, please include the date of censoring and the date up to which the data on mortality was complete.

Discussion

The main limitation of this study is the unavailability of causes of death for the matched controls. Consequently, comparing mortality after breast cancer diagnosis between survivors and controls is not informative, as survivors have a higher prevalence of comorbidities due to previous cancer treatments. To determine whether breast cancer therapeutic modifications and treatment-related toxicities contribute to excess mortality in survivors, the authors should have examined cause-specific mortality, specifically deaths from breast cancer. I understand that this data was not available for controls, so this significant limitation should be thoroughly addressed in the discussion.

If possible, please present the results (HRs) from Table 2 as Forest Plots.

Version 1:

Reviewer comments:

Reviewer #1

(Remarks to the Author)

I applaud the authors for their in-depth response to the comments and corresponding modifications to the manuscript. I believe the manuscript is acceptable for publication in its current form.

Reviewer #2

(Remarks to the Author)

The authors made a reasonable attempt to rectify the concerns of this reviewer.

Reviewer #3

(Remarks to the Author)

Thank you for providing the revised version of the manuscript entitled "Treatment, toxicity, and mortality after subsequent breast cancer in female survivors of childhood cancer." I have reviewed the changes made in response to the comments provided by myself and the other reviewers.

I am satisfied with the authors' response and the revisions made to the manuscript. My concerns have been adequately addressed.

RESPONSE TO REVIEWERS

Re: NCOMMS-24-49443-T

Revised manuscript title: "Treatment, toxicity, and mortality after subsequent breast cancer in female survivors of childhood cancer"

We thank the *Nature Communications* reviewers for their careful review of our manuscript. We have responded to all points made by the reviewers by including clarifying edits or by providing additional analyses. These changes have led to a substantially improved manuscript. In the following pages, we provide a point-by-point response for each of the three reviewers.

Reviewer #1 (Remarks to the Author):

In their study, Im et al describe therapeutics, treatment-related toxicities and mortality in 431 female childhood cancer survivors from the Childhood Cancer Survivor Study (CCSS) who later developed breast cancer (“survivors”) and compare treatment and survival characteristics in these patients to matched females with primary adult breast cancer (“controls”) and to population-based SEER rates. They found that survivors are often treated differently for their breast cancer than controls and that survivors have increased mortality compared to controls, both overall and stratified by breast cancer characteristics. The results are interesting and have the potential to advance the science, but the manuscript suffers from lack of focus and flow and from data missingness, making interpretation of results very challenging.

Response: We thank the reviewer for their comments. We have carefully reviewed the manuscript and made substantial revisions to present all results in a linear fashion: analyses conducted in childhood cancer survivors are now presented first, followed by analyses comparing survivors and females with first primary breast cancer (controls). These edits have improved the focus and flow of the manuscript. To address the reviewer’s comments about data missingness, we have added a CONSORT diagram with the beginning sample size, exclusions, and event numbers included in each of the primary analyses (**new Supplementary Figure 1, please see p. 11**). We have also made revisions throughout the manuscript that clarify the implications of missingness on interpreting our results.

Major comments

1. The manuscript is extremely dense with many analyses being carried out, making it challenging to follow. Moreover, the results section occasionally moves back in forth between tables and figures in a somewhat random pattern. For instance, in the “Treatment Selection, modification and toxicity section”, the authors refer to Supplementary Table 4, then Supplementary Table 5, then Supplementary Figure 1 and then back again to Supplementary Table 4 when comparing survivors to matched controls. Following this, the last sentence in the same paragraph refers to Supplementary Table 6, which does not include the matched controls. This back and forth severely compromises the readability. The manuscript would benefit from a more focused and linear approach.

Response: We thank the reviewer for making these helpful points. To address this comment, we reorganized the manuscript. All tables and figures now follow in linear fashion. The revised version first presents results from analyses of subsequent breast cancer conducted in childhood cancer survivors, followed by comparisons of treatment and mortality among survivors and controls. Therefore, the following four subheadings are used to organize the Results section:

1. Subsequent breast cancers in survivors of childhood cancer
2. Mortality after subsequent breast cancer
3. Treatment for first primary versus subsequent breast cancer
4. Mortality after first primary versus subsequent breast cancer

To address the reviewer’s comment about the density of our manuscript, we removed results that were not absolutely necessary for explaining our findings. These include previously enumerated Supplementary Figures 2-3, and Supplementary Tables 5, 10, 11, and 12.

We kindly refer the reviewer to look at highlighted, revised manuscript because the specific edits are too numerous to include in this point-by-point response.

2. Keeping with the theme of the comment above: Supplementary Tables 1 and 2 are among the survivors only, Supplemental Tables 3-5 include both survivors and matched controls, Supplemental Tables 6-8 are among the survivors only, Supplemental Tables 9-12 include both survivors and matched controls, Supplemental Tables 13-14 are among the survivors only, Supplemental Tables 15 and 16 include both survivors and matched controls, Supplementary Table 17 compares survivors to SEER data, and finally Supplementary Table 18 includes both survivors and matched controls. This whiplash approach makes the manuscript far too complicated to follow.

Response: The reviewer's suggestion to linearly organize our results was helpful and constructive. The supplementary tables are now reordered as follows:

- Supplementary Tables 1-7 now summarize analyses conducted in survivors only;
 - Supplementary Tables 8-13 summarize treatment, toxicity, and mortality analyses comparing survivors and controls; and
 - Supplementary Table 14 compares mortality among females with primary breast cancer in SEER with controls in the current study.
3. More detailed information on the matching approach should be provided in the supplementary material. I'm unsure what they authors mean by "ordered by priority." Does this mean that not all matching variables were used for each match?

Response: The reviewer is correct, matching was imperfect. However, the distributions of matching variables were very similar between cases and matched controls, with the exception of race and ethnicity (Supplementary Table 8, provided below). Following our primary analysis, we conducted a sensitivity analysis to evaluate the potential for residual confounding after matching and estimated mortality hazard ratios using models that adjusted for breast cancer diagnosis age, diagnosis year, race and ethnicity, and if appropriate, disease histology (Supplementary Table 12, provided below). This sensitivity analysis showed similar results to our primary analysis. Clarifying edits to the Methods and Discussion have been made to address this point (please see below).

Edits to Methods (edits are highlighted): A total of 344 females diagnosed with primary LCIS/DCIS (n=85) or invasive (n=259) breast cancer were matched one-to-one to CCSS participants by breast cancer diagnosis age, diagnosis year, and disease histology and/or stage. Because widespread testing for estrogen receptor status only began in the mid-1990s, hormone receptor and HER2 status were not consistently available for participants diagnosed before 1995. Therefore, when such data were available, controls were also matched on hormone receptor status and HER2 status. Race and ethnicity was also matched when multiple controls could be matched on breast cancer disease characteristics. When multiple suitable matches considering all matching factors were available, the diagnosis date closest to that of the survivor case was selected...

...We also performed analyses comparing survivors with subsequent breast cancer and matched controls with first primary breast cancer... A sensitivity analysis to assess potential residual confounding after matching was conducted, i.e., evaluating mortality hazard ratios also adjusted for breast cancer diagnosis age and diagnosis year, histology, and race/ethnicity, along with receipt of guideline-concordant breast cancer treatment.

Edits to Discussion (edits are highlighted): The procedure to match females with first primary breast cancer to CCSS survivors was imperfect, leading to the inclusion of fewer White matched controls. Sensitivity analyses where we also adjusted for matching variables led to similar results.... Because Black females with breast cancer have consistently worse

survival^{1,2} and also constituted the largest subgroup of non-White matched controls..., the excess mortality reported here may be underestimated.

Supplementary Table 8: Characteristics among CCSS survivors and general population controls

	Survivors (N=344)		General population (N=344)	
	N	% or median (IQR)	N	% or median (IQR)
Age at first breast cancer diagnosis Median (IQR)	344	41 (36-45)	344	42 (37-47)
Treatment year				
Before 2000	63	18.2	56	16.3
2000-2009	157	45.3	146	42.4
2010-2016	124	36.5	142	41.3
Race ^a				
White	327	95.1	237	78.0
Non-White	17	4.9	67	22.0
Black	13	3.8	48	15.8
Asian/Pacific Islander	2	0.6	13	4.3
Other	2	0.6	6	2.0
Stage of first breast cancer ^b				
0	81	24.1	81	24.0
I	111	33.1	113	33.4
II	100	30.6	99	29.3
III	31	9.2	36	10.7
IV	10	3.0	9	2.7
Histology				
DCIS/LCIS only	85	24.5	85	24.7
Invasive	259	75.5	259	75.3
Progesterone receptor status ^c				
Positive	197	69.2	180	65.5
Estrogen receptor status ^c				
Positive	233	78.6	209	75.2
HER2 status ^c				
Positive	59	25.9	57	24.1
Vital status ^d				
Alive	228	66.5	208	86.3
Dead	116	33.5	33	13.7

Abbreviations: IQR, interquartile range; DCIS/LCIS, ductal carcinoma in situ/lobular carcinoma in situ; HER2, human epidermal growth factor receptor 2.

- Missing race in 40 controls.
- Missing breast cancer stage in 11 survivors and 6 controls.
- Missing breast cancer hormone receptor status and HER2 status, as follows: progesterone receptor status (62 survivors, 69 controls), estrogen receptor status (47 survivors, 66 controls); HER2 status (119 survivors, 107 controls).
- Missing vital status for 103 controls.

Supplementary Table 12: All-cause mortality hazard ratios comparing survivors to matched controls, adjusting for key matching variables

Subgroup	HR (95% CI)	P
Entire sample	4.14 (2.44-7.00)	<0.001
LCIS/DCIS	9.77 (2.04-46.79)	<0.01
Invasive	3.45 (1.96-6.07)	<0.001
Mastectomy, no radiotherapy	14.70 (3.58-60.42)	<0.001
Lumpectomy, with radiotherapy	7.77 (2.00-30.20)	<0.01
Neither treatment group	1.73 (0.84-3.55)	0.14

Abbreviations: DCIS/LCIS, ductal carcinoma in situ/lobular carcinoma in situ; HR, hazard ratio; CI, confidence interval; P, p-value. All models adjusted for receipt of guideline-discordant care and the following matching variables: first breast cancer diagnosis age and year; race/ethnicity; breast cancer histology (except in LCIS/DCIS and invasive subgroups). Mastectomy refers to unilateral or bilateral mastectomy.

- The authors state in the statistical methods that conditional logistic regression analysis was used to compare survivors to matched primary cancer controls, sometimes adjusting for

other characteristics and sometimes not. I could not determine in what specific adjustments were made in each of the numerous tables and figures comparing survivors to their matched controls.

Response: We have reviewed the manuscript tables and figures to assure that all model information comparing survivors to controls are available in table or figure titles and/or footnotes. These are limited to Figures 3 and 4, and Supplementary Tables 9, 10, 12, and 13. We have also revised the Methods section to clarify all regression model details; for convenience, we have annotated this section with corresponding figure and table numbers in italics (please see below).

Edits to Methods (edits are highlighted; corresponding tables/figures are annotated in italics):

Statistical analysis

...We also performed analyses comparing survivors with subsequent breast cancer and matched controls with first primary breast cancer. Univariate conditional logistic regression models evaluated differences in prescribed breast cancer treatment modalities conditioned on matched pairs [*Figure 3, Supplementary Table 9*]. Odds ratios for breast cancer treatment modifications and acute toxicities comparing survivors with matched controls were obtained, specifically among those treated with chemotherapy and prescribed guideline-concordant treatment, using logistic regression models adjusted for breast cancer diagnosis age, year, histology and race/ethnicity [*Supplementary Table 10*]. Cumulative all-cause mortality probabilities were estimated with the Kaplan-Meier method and log-rank tests with robust variant estimation³ compared cumulative incidence curves. Cox regression models with robust variance estimation³ were used to compare mortality hazard rates between survivors and controls, adjusting for receipt of guideline-concordant breast cancer treatment [*Figure 4*]. A sensitivity analysis to assess potential residual confounding after matching was conducted, i.e., evaluating mortality hazard ratios also adjusted for breast cancer diagnosis age and diagnosis year, histology, and race/ethnicity, along with receipt of guideline-concordant breast cancer treatment [*Supplementary Table 12*]. This multivariable model was then further adjusted for experiencing any treatment modification or acute toxicity to assess whether these factors attenuated differences in mortality risk between survivors and controls [*Supplementary Table 13*].

5. Table 2 (“Risk factors for all-cause, health-related cause, and breast cancer-specific mortality in childhood cancer survivors”) reports 99 deaths overall, but only 43 for non-breast cancer related mortality and 33 for breast cancer related mortality. This only sums to 76 deaths. Please verify that these numbers are correct and presumed missingness was based in inability to ascertain cause of death. If this is indeed true, how did the authors account for missing cause of death when evaluating associations with cause-specific mortality?

Response: In the revision process, we found that we previously provided an outdated version of Table 2 in our manuscript. We apologize for any confusion this may have caused. The correct display now includes updated sample and event numbers. These corrections did not change any conclusions related to these results. All related revisions to the Results are highlighted in the manuscript and provided below.

In the results presented in Table 2 (now Figure 2; HRs and 95% CIs are presented as forest plots per the request of another reviewer), all Cox proportional hazards regression models used complete case data including 314 survivors. In this subgroup, 97 survivors died. Cause of death information was available for 80 out of the 97 deaths, where 30 died of

breast cancer, 44 died of other health-related causes, and 6 died of external causes. Cause-specific mortality HRs were obtained from Cox proportional hazards regression models where deaths that were not cause-specific (i.e., breast cancer or other health-related causes) led to the removal of affected subjects from the at-risk pool and survivors with missing cause of death information were excluded from these analyses. We have provided clarifying edits to the Methods (below).

Edits to Methods (edits are highlighted): Cox proportional hazards models for all-cause and cause-specific mortality using age as the time scale⁴ were used to estimate hazard ratios (HRs) for guideline-discordant breast cancer treatment and childhood cancer treatments (continuous chest radiotherapy and anthracycline doses with one degree-of-freedom terms), adjusting for breast cancer histology and diagnosis year, breast cancer treatments (any radiotherapy; any anthracyclines), and time-varying non-breast SMNs and major cardiovascular late effects. For cause-specific mortality HRs, survivors without cause of death information were excluded and deaths that were not cause-specific led to the removal of affected subjects from the at-risk pool.

Edits to Results (edits are highlighted): Multivariable models including survivors with complete clinical information assessed all-cause and cause-specific mortality risk factors (n=314, 97 deaths; Figure 2). Most survivors who were omitted from these analyses were excluded due to insufficient breast cancer treatment information (69%; Supplementary Figure 1). We found being prescribed guideline-discordant breast cancer care (HR=3.04, 95% CI: 1.62-5.71), major cardiovascular events (HR=1.94, 95% CI: 1.17-3.20), non-breast SMNs (HR=2.14, 95% CI: 1.17-3.92), and subsequent breast irradiation (HR=2.34, 95% CI: 1.38-3.97) were risk factors for all-cause mortality. We observed being prescribed guideline-discordant care had an adjusted 7.17-fold greater risk of breast cancer-related death (95% CI: 3.05-16.86). Non-breast SMNs (HR=2.98, 95% CI: 1.21-7.33), childhood chest RT dose (per 10 Gy, HR=1.42, 95% CI: 1.14-1.77), and subsequent breast irradiation (HR=2.33, 95% CI: 1.09-5.01) were risk factors for other health cause-related death.

Figure 2: Risk factors for all-cause and cause-specific mortality after subsequent breast cancer in childhood cancer survivors. Abbreviations: BC, breast cancer; RT, radiation therapy; CV, cardiovascular; SMN, subsequent malignant neoplasm. HRs and corresponding 95% confidence intervals (CIs) from multivariable Cox proportional hazards regression models evaluating associations between primary and breast cancer treatments and the mortality hazard rate using age as the time scale and adjusting for covariates included above are shown, as well as breast cancer diagnosis year and histology (invasive versus non-invasive) or stage (III/IV versus I/II, invasive carcinomas only). HRs and 95% CIs are annotated for statistically significant variables (two-sided $P < 0.05$). Primary cancer chest RT dose is provided per 10 Gray and primary anthracycline dose is provided per 100 mg/m².

6. Table 2 includes continuous effects (breast cancer diagnosis year, chest RT dose, anthracycline dose): I assume these associations were fit using a one degree-of-freedom term? If so, this should be noted in the statistical methods. Modeling continuous effects in this manner assumes a log-linear dose-response association between exposure and risk of death. Did the authors examine the functional form of association to ensure that modeling assumptions were met? Also, how did the authors model subjects who did not receive anthracycline or chest RT? Were they given a value of 0? I could imagine a scenario where risk or death increases dramatically in patients not receiving any anthracycline or chest RT to those receiving one dose, with more modest increases as dosage increases, which would be counter to the log-linear association modeling assumption.

Response: In this analysis of survivors, we clarify that RT or anthracycline doses to treat breast cancer were treated dichotomously (any versus no exposure) because of dose data missingness. As noted in our Methods, dosages of treatments for primary childhood cancer reflect delivered doses (i.e., this includes zero doses): we used maximum cumulative chest RT doses based on dosimetry and maximum cumulative doxorubicin-equivalent doses of anthracyclines delivered within 5 years of their primary cancer diagnosis.

We previously only observed associations between risk of other health-related causes and chest RT dose for primary childhood cancer among survivors with subsequent breast cancer. To assess whether adjusting for continuous primary cancer treatment dose is appropriate, we evaluated dichotomous (any versus no exposure) and categorical doses (lower and higher doses versus none, as defined in the table below) for primary cancer chest RT and anthracyclines. These results do not show evidence of increasingly more modest changes in risk with increasing chest RT dose or any appreciable changes in regression coefficients for other variables, indicating our use of continuous primary cancer treatment dose is appropriate. We have edited the methods to clarify how these variables were treated (please see below).

Childhood cancer treatment dose	Variables	HR (95% CI)	P
Continuous	Experienced major CV event ^b	1.81 (0.86-3.84)	0.12
	Experienced non-breast SMN ^b	2.98 (1.21-7.33)	0.018
	Guideline-discordant BC treatment prescribed	1.84 (0.57-5.97)	0.31
	Primary: Chest RT dose (per 10 Gy)	1.42 (1.14-1.77)	<0.01
	Primary: Anthracycline dose (per 100 mg/m ²)	1.14 (0.91-1.41)	0.25
	Breast cancer: Any anthracyclines	1.42 (0.73-2.76)	0.30
	Breast cancer: Any radiation	2.33 (1.09-5.01)	0.029
Dichotomous	Experienced major CV event ^b	1.94 (0.93-4.04)	0.078
	Experienced non-breast SMN ^b	3.37 (1.39-8.18)	<0.01
	Guideline-discordant BC treatment prescribed	2.12 (0.70-6.40)	0.18
	Primary: Any chest RT	5.83 (1.75-19.39)	<0.01
	Primary: Any anthracyclines	2.00 (0.90-4.44)	0.088
	Breast cancer: Any anthracyclines	1.51 (0.76-3.00)	0.24
	Breast cancer: Any radiation	2.29 (1.08-4.85)	0.031
Categorical	Experienced major CV event ^b	1.93 (0.92-4.04)	0.081
	Experienced non-breast SMN ^b	3.31 (1.36-8.02)	<0.01
	Guideline-discordant BC treatment prescribed	2.25 (0.72-7.09)	0.16
	Primary: Chest RT dose, >0 to ≤35 Gy	5.23 (1.47-18.60)	0.011
	Primary: Chest RT dose, >35 Gy	6.07 (1.77-20.89)	<0.01
	Primary: Anthracycline dose, >0 to ≤250 mg/m ²	2.53 (0.88-7.32)	0.086
	Primary: Anthracycline dose, >250 mg/m ²	1.83 (0.70-4.78)	0.22
	Breast cancer: Any anthracyclines	1.55 (0.77-3.12)	0.22
Breast cancer: Any radiation	2.34 (1.09-5.02)	0.030	

Edits to Methods (edits are highlighted): Cox proportional hazards models for all-cause and cause-specific mortality using age as the time scale⁴ were used to estimate hazard ratios (HRs) for guideline-discordant breast cancer treatment and childhood cancer treatments (continuous chest radiotherapy and anthracycline doses with one degree-of-freedom terms), adjusting for breast cancer histology and diagnosis year, breast cancer treatments (any radiotherapy; any anthracyclines), and time-varying non-breast SMNs and major cardiovascular late effects.

7. The authors report that 164 of the 431 CCSS subjects subsequently died, but Table 2 suggests that only 320 subjects were used in the mortality analyses, among whom 99 died. My guess is that the discrepancy is because of dropout due to missing values for some of the variables of interest, but the authors need to provide more details. Moreover, the death rate differs among those included and not included in Table 2: of the 320 included in Table 2, 99 subsequently died (31%), whereas among those not included in Table 2, 59% ((164-99)/(431-320)) subsequently died. This raises serious concerns about the representativeness of the subjects used in Table 2 analyses.

Response: We agree with the reviewer that data missingness is an important concern. In Figure 2 (previously Table 2), we present results from complete case analyses in a subgroup with 314 survivors, among whom 97 died. The majority of the 117 excluded survivors (69% or 81 survivors) had incomplete breast cancer treatment documentation, precluding assessment of whether care was prescribed following national care guidelines for 75 survivors. This data missingness is likely related to the age of their treatment records: 47% of those excluded were treated before 2000. The 15-year mortality probability after breast cancer diagnosis was higher among excluded survivors (54%, 95% CI: 43-63%).

However, the survivors included in analyses presented in Table 2 are representative of the survivors included in the mortality analysis with matched controls. We found high overlap (78%) between survivors that were evaluated in the mortality analysis with matched controls (Figure 4) and also included in analyses presented in Figure 2. In the matched analyses, 16-18% were treated before 2000 (Supplementary Table 8). The 15-year overall mortality probability after breast cancer diagnosis in the subgroup of 314 survivors included in analyses presented in Figure 2 was 41% (95% CI: 35-47%), which is identical to the 241 survivors included in the mortality analysis with matched controls (41%; 95% CI: 32-49%; Figure 4 and Supplementary Table 11).

Overall, a major factor underlying data missingness in the mortality risk analyses is related to insufficient breast cancer treatment documentation, frequently excluding survivors treated during earlier time periods and with a higher rate of death. This would suggest that this data missingness could lead to reported mortality risk associations that are conservative.

Edits to Results (edits are highlighted): Multivariable models including survivors with complete clinical information assessed all-cause and cause-specific mortality risk factors (n=314, 97 deaths; Figure 2). Most survivors who were omitted from these analyses were excluded due to insufficient breast cancer treatment information (69%; Supplementary Figure 1)...

Edits to Discussion (edits are highlighted): Study limitations must be acknowledged... These data are also limited by missingness. For example, in multivariable analyses of mortality risk factors among survivors with subsequent breast cancer (Figure 2), 117 survivors were excluded due to data missingness. Nearly half of those excluded were treated for breast cancer before 2000; accordingly, the most common reason for exclusion was related to

incomplete breast cancer treatment documentation. However, the survivors included in this mortality risk factor analysis had high overlap with and are representative of the survivors assessed in mortality analyses with matched controls, whereas excluded survivors had higher mortality (54%, 95% CI: 43-63%). Because... excluded survivors had a higher rate of death, the excess mortality reported here may be underestimated.

8. This missingness issue extends to other tables and figures as well. The authors note this as a limitation in the discussion section, but I'm concerned it may have a profound effect on the results that has the potential to alter interpretation of results.

Response: We have made substantial edits to the manuscript to clarify the potential impact of data missingness on the conclusions of this study. These include the addition of a new CONSORT diagram with the beginning sample size, exclusions, and event numbers included in each of the primary analyses (please see our response to point 9 below) and edits to the Discussion (provided below).

Overall, data missingness is an important limitation, but is unlikely to change the overall conclusions drawn in this study. Data missingness related to breast cancer treatments was similar in survivors and controls (Supplementary Table 9), and prescribed breast cancer treatments were highly consistent with the national clinical practice guidelines in both groups ($\geq 93\%$). This suggests treatment data missingness is likely non-differential. Data missingness related to analyses of mortality risk is the more concerning threat. Given the inclusion of more Black female primary breast cancer patients among controls and the higher mortality rate among survivors excluded from analyses, our overall conclusions remain the same (please see Discussion below; we apologize for the overlap with previous responses above, but provide this information again for the reviewer's convenience).

Edits to Discussion (edits are highlighted): ...These data are also limited by missingness. For example, in multivariable analyses of mortality risk factors among survivors with subsequent breast cancer (Figure 2), 117 survivors were excluded due to data missingness. Nearly half of those excluded were treated for breast cancer before 2000; accordingly, the most common reason for exclusion was related to incomplete breast cancer treatment documentation. However, the survivors included in this mortality risk factor analysis had high overlap with and are representative of the survivors assessed in mortality analyses with matched controls, whereas excluded survivors had higher mortality (54%, 95% CI: 43-63%). Because Black females with breast cancer have consistently worse survival^{1,2} and also constituted the largest subgroup of non-White matched controls and excluded survivors had a higher rate of death, the excess mortality reported here may be underestimated...

9. The manuscript would greatly benefit from a CONSORT diagram that displays the beginning sample size, exclusions, and number included in each of the analyses presented in the tables and figures.

Response: We agree with the reviewer and have added a new CONSORT diagram with the beginning sample size, exclusions, and event numbers included in each of the major analyses (please see the new Supplementary Figure 1 below). Related manuscript edits are provided below.

Edits to Results (edits are highlighted): ...Among 11,550 females participating in CCSS, 431 had a pathology-ascertained subsequent breast tumor (in situ or invasive) during adulthood (age ≥ 18 years) occurring at least five years after primary cancer diagnosis (Supplementary Figure 1)...

...Multivariable models including survivors with complete clinical information assessed all-cause and cause-specific mortality risk factors (n=314, 97 deaths; Figure 2). Most survivors who were omitted from these analyses were excluded due to insufficient breast cancer treatment information (69%; Supplementary Figure 1)...

Supplementary Figure 1: Overview of participant inclusions and exclusions across all primary analyses.

Supplementary Table 7 (corresponds with Figure 1): Breast cancer- and other cause-specific mortality probabilities among CCSS survivors, stratified by disease histology, guideline-concordant breast cancer treatment receipt, and treatment modality

	N	Deaths	5 years		10 years		15 years	
			Breast cancer	Other cause	Breast cancer	Other cause	Breast cancer	Other cause
			Probability (95% CI)					
Overall	402	BC: 61 Other: 74	0.08 (0.06-0.12)	0.03 (0.02-0.05)	0.14 (0.11-0.18)	0.11 (0.08-0.15)	0.19 (0.15-0.24)	0.19 (0.15-0.25)
DCIS/LCIS	98	BC: 4 Other: 20	0.01 (0.00-0.07)	0.03 (0.01-0.10)	0.03 (0.01-0.10)	0.14 (0.08-0.24)	0.05 (0.02-0.14)	0.22 (0.14-0.36)
Invasive	300	BC: 57 Other: 53	0.11 (0.08-0.15)	0.03 (0.01-0.05)	0.17 (0.13-0.22)	0.10 (0.07-0.14)	0.23 (0.18-0.29)	0.18 (0.14-0.25)
Guideline-concordant care	310	BC: 32 Other: 51	0.04 (0.02-0.07)	0.03 (0.02-0.06)	0.09 (0.06-0.13)	0.11 (0.08-0.15)	0.14 (0.10-0.20)	0.17 (0.13-0.24)
Guideline-discordant care	23	BC: 9 Other: 4	0.30 (0.16-0.56)	0	0.35 (0.20-0.62)	0.16 (0.06-0.45)	0.43 (0.26-0.73)	0.26 (0.11-0.63)
Lumpectomy, RT	42	BC: 3 Other: 5	0	0	0.07 (0.02-0.26)	0.07 (0.02-0.29)	0.12 (0.04-0.37)	0.25 (0.11-0.56)
Mastectomy, no RT	239	BC: 21 Other: 47	0.03 (0.01-0.06)	0.02 (0.01-0.05)	0.07 (0.04-0.12)	0.11 (0.08-0.17)	0.12 (0.08-0.18)	0.19 (0.14-0.26)
Neither	84	BC: 19 Other: 15	0.19 (0.12-0.3)	0.05 (0.02-0.13)	0.24 (0.16-0.36)	0.11 (0.06-0.21)	0.24 (0.16-0.36)	0.20 (0.12-0.34)

Abbreviations: DCIS/LCIS, ductal carcinoma in situ/lobular carcinoma in situ; BC, breast cancer; RT, radiation therapy. Mastectomy refers to unilateral or bilateral mastectomy.

Supplementary Table 11 (corresponds with Figure 4): All-cause mortality probabilities in CCSS survivors and matched controls, stratified by breast cancer histology and treatment modality

	Entire sample		LCIS/DCIS		Invasive	
	Controls (n=241, 33 deaths)	Survivors (n=241, 84 deaths)	Controls (n=63, 3 deaths)	Survivors (n=63, 16 deaths)	Controls (n=178, 30 deaths)	Survivors (n=178, 68 deaths)
5-year	0.06 (0.03-0.09)	0.13 (0.08-0.17)	0.02 (0.00-0.05)	0.03 (0.00-0.08)	0.08 (0.04-0.12)	0.16 (0.10-0.21)
10-year	0.08 (0.05-0.11)	0.29 (0.22-0.35)	0.02 (0.00-0.05)	0.17 (0.05-0.27)	0.10 (0.06-0.15)	0.33 (0.25-0.41)
15-year	0.14 (0.09-0.19)	0.41 (0.32-0.49)	0.02 (0.00-0.05)	0.39 (0.19-0.54)	0.18 (0.11-0.25)	0.42 (0.32-0.50)
	Mastectomy*, no RT		Lumpectomy, RT		Neither treatment group	
	Controls (n=80, 4 deaths)	Survivors (n=147, 44 deaths)	Controls (n=94, 7 deaths)	Survivors (n=29, 8 deaths)	Controls (n=61, 20 deaths)	Survivors (n=53, 23 deaths)
5-year	0.02 (0.00-0.06)	0.05 (0.02-0.09)	0.01 (0.00-0.03)	0.04 (0.00-0.11)	0.20 (0.09-0.29)	0.28 (0.14-0.39)
10-year	0.02 (0.00-0.06)	0.23 (0.15-0.30)	0.02 (0.00-0.05)	0.20 (0.00-0.35)	0.23 (0.12-0.33)	0.43 (0.26-0.56)
15-year	0.02 (0.00-0.06)	0.32 (0.21-0.41)	0.08 (0.01-0.14)	0.49 (0.16-0.69)	0.39 (0.22-0.52)	0.51 (0.31-0.66)

Abbreviations: DCIS/LCIS, ductal carcinoma in situ/lobular carcinoma in situ; RT, radiation therapy. *Mastectomy refers to unilateral or bilateral mastectomy.

Minor comments

- Supplementary Table 3. Even though the authors attempted to match on race/ethnicity, there seem to be large differences in race between the survivors and the matched general population subjects. The authors should acknowledge this in the text of their manuscript.

Response: We have addressed this reviewer’s point in an earlier response (please see our response to point #3). We provide the same manuscript edits here for the reviewer’s convenience.

Edits to Methods (edits are highlighted): ...We also performed analyses comparing survivors with subsequent breast cancer and matched controls with first primary breast cancer... A sensitivity analysis to assess potential residual confounding after matching was conducted, i.e., evaluating mortality hazard ratios also adjusted for breast cancer diagnosis age and diagnosis year, histology, and race/ethnicity, along with receipt of guideline-concordant breast cancer treatment.

Edits to Discussion (edits are highlighted): The procedure to match females with first primary breast cancer to CCSS survivors was imperfect, leading to the inclusion of fewer White matched controls. Sensitivity analyses where we also adjusted for matching variables led to similar results.... Because Black females with breast cancer have consistently worse survival^{1,2} and also constituted the largest subgroup of non-White matched controls..., the excess mortality reported here may be underestimated.

- The authors should justify why they are including LCIS cases in addition to DCIS and invasive breast cancer.

Response: We thank the reviewer for raising this point. In situ subsequent breast cancers were included given previous evidence indicating their increased risk of mortality. For completeness, we also provide results for subgroups with LCIS/DCIS and invasive breast cancers separately.

Edits to Discussion (edits are highlighted): Including female survivors with in situ subsequent breast cancers affect estimates of excess mortality. These were included given previous

evidence indicating their increased risk of mortality especially for non-breast cancer-related causes.⁵ We conducted analyses limited to and excluding those with in situ carcinomas; notably, we observed conditional excess mortality persists after excluding in situ carcinomas.

Reviewer #2 (Remarks to the Author):

12. The comparison group for the CCSS was a multi-institutional group of females with primary breast cancer diagnosed at three academic medical centers. Do the authors believe that the outcomes of breast cancer at top academic medical centers in the US are slightly better than the overall national average? If yes, then the difference between the control and experimental group would be higher than expected. Please give national stats for the breast cancer outcomes. The 1:1 match is well noted.

Response: We thank the reviewer for raising this point. We have edited the Discussion (please see below) accordingly. However, we note that we observed comparable crude survival estimates in SEER and among matched controls, which further suggests that our mortality risk estimates are reasonable.

Edits to Discussion (edits are highlighted): We also observed the crude survival estimates in SEER and matched controls in these data are largely consistent, especially for situ breast cancer... Because Black females with breast cancer have consistently worse survival^{1,2} and also constituted the largest subgroup of non-White matched controls and excluded survivors had a higher rate of death, the excess mortality reported here may be underestimated. This bias may be tempered by the use of controls drawn from US academic medical centers which typically have lower cancer mortality rates than community-based hospitals.^{6,7}

13. The external validity of the study is poor, since the long term survivors do not fare as well as in the US (compared to developing countries). Please add a section on this in point in the discussion

Response: We agree with the reviewer that the generalizability of our findings are limited to childhood cancer survivors from the US and Canada and have edited the Discussion (please see below).

Edits to Discussion (edits are highlighted): Because these data reflect the experience of five-year childhood cancer survivors with subsequent breast cancer and females with first primary breast cancer treated in the US and Canada, the generalizability of our findings to other countries is limited.

14. Some of the risk factors are not described. In fact, <25% of the overall breast cancers may be attributed to inheritance in some populations (e.g. BRCA), and the genome is mentioned in the manuscript. However, majority of the breast cancers are due to epigenetic changes. We have abundant published data that certain common risk factors increase the risk of breast Ca - alcohol, fast food, obesity, (smoking association is weaker), early menarche, late menopause, absence of breast feeding, having kids at a very late age etc. We also have abundant data on BMT and cancer survivors showing multiple psychological comorbidities. Thus the most important question is that whether the risk of increased breast cancer is due to cancer treatments, or actually due to behavioural changes due to cancer in these survivors, or both? Please explain how can we exclude confounders and effect modifiers in the absence of data present on epigenetic risk factors.

Response: The reviewer raises important points on important possible breast cancer risk factors. We clarify that an assessment of the multiple factors that may be associated with

increased risks of subsequent breast cancer among female survivors of childhood cancer is beyond the scope of this manuscript. There is abundant literature in this area and is referenced in the Introduction section of our manuscript. The focus of this manuscript is to characterize differences in prescribed breast cancer treatments between survivors and females with first primary breast cancer and evaluate whether survivors' therapeutic tradeoffs impact mortality risk. We have reviewed our Discussion section and made clarifying revisions to address this point (please see below).

Edits to Discussion (edits are highlighted): Genetic susceptibility for breast cancer and its impact on survival was not assessed. Despite the potential for inferior outcomes among younger females with first primary breast cancer⁸⁻¹⁰ or related to genetic breast cancer predisposition¹¹, these results reflect analyses where survivors and controls were matched based on diagnosis age, disease stage, and if available, molecular subtype. Our analyses also did not account for all hypothesized prognostic predictors, e.g., novel biomarkers, lifestyle and reproductive factors, given their limited availability across cases and controls, and instead limited our analyses to consider classical clinical and pathological features predictive of survival, including histology, stage, and molecular subtype.

Reviewer #3 (Remarks to the Author):

Thank you for inviting me to review the manuscript titled "Treatment, Toxicity, and Mortality After Primary Versus Subsequent Breast Cancer: A Report from the Childhood Cancer Survivor Study." The authors have explored a significant research question. I have several major and minor comments that should be addressed to enhance the manuscript's clarity and quality.

Abstract, Page 3

15. The abstract needs some rewording to be more informative and clearer.

Response: We thank the reviewer for their suggestion. We have updated the abstract to be more informative and clear (full abstract provided below). This version has been edited to meet the journal's editorial requirements (e.g., 150-word limit).

Updated abstract: Childhood cancer survivors, particularly those who received chest radiotherapy, are at high risk for developing subsequent breast cancer. Minimizing long-term toxicity risks associated with additional radiotherapy and chemotherapy is a priority, but therapeutic tradeoffs have not been comprehensively characterized and their impact on survival is unknown. In this study, 431 female childhood cancer survivors with subsequent breast cancer from a multicenter retrospective cohort study were evaluated. Compared with one-to-one matched females with first primary breast cancer, survivors were as likely to be prescribed guideline-concordant treatment (N=344 pairs; survivors: 94%, controls: 93%), but more frequently underwent mastectomy (survivors: 81%, controls: 60%) and were less likely to be treated with anthracyclines (survivors: 47%, controls: 66%) or radiotherapy (survivors: 18%, controls: 61%). Despite this, survivors had nearly 3.5-fold (95% CI: 2.17-5.57) greater mortality risk. Here, we show survivors with subsequent breast cancer face excess mortality despite therapeutic tradeoffs and require specialized treatment guidelines.

16. Background, line 64: If the wording allows, please add 'particularly those who received chest radiation' after 'Childhood cancer survivors.' So, the sentence would read as follows: "Childhood cancer survivors, particularly those who received chest radiation, are at high risk for developing subsequent breast cancer with higher mortality than females with first primary breast cancer."

Response: We have edited the abstract as suggested by the reviewer, but shortened the sentence to meet the 150-word limit: "Childhood cancer survivors, particularly those who received chest radiotherapy, are at high risk for developing subsequent breast cancer."

17. Methods, line 67: Please mention the number of female five-year survivors of childhood cancer identified from the multicentre retrospective cohort study, as well as the number of controls (females with first primary breast cancer).

Response: We have edited the abstract as suggested by the reviewer: "In this study, 431 female childhood cancer survivors with subsequent breast cancer from a multicenter retrospective cohort study were evaluated. Compared with one-to-one matched females with first primary breast cancer, survivors were as likely to be prescribed guideline-concordant treatment (N=344 pairs; survivors: 94%, controls: 93%)..."

18. Line 69: For clarity, please add 'first' before 'primary.' So, the sentence would read as: "and breast cancer characteristics with females with first primary breast cancer..."

Response: We have edited the abstract as suggested by the reviewer: “Compared with one-to-one matched females with first primary breast cancer, survivors were as likely...”

19. Line 72: As mentioned above, add 'first' before 'primary breast cancer.' Please correct this throughout the manuscript, so it's clear to the readers that it was the first cancer diagnosed in controls.

Response: We have edited the all manuscript materials to indicate controls are females with “first primary breast cancer”. These changes are highlighted in the main text.

20. Line 79: How many deaths were among controls?

Response: We apologize for not providing this information in the abstract. However, due to 150-word limit, we have included this information at the first opportunity in the Results. Please see the corresponding edit below.

Edits to Results (edits are highlighted): In analyses with matched controls with medical record- or registry-ascertained mortality data (n=241 pairs with 84 deaths among survivors and 33 deaths among controls), the 15-year mortality probability after breast cancer diagnosis was 41% among survivors (95% CI: 32-49%) versus 14% in controls (95% CI: 9-19%)...

21. Introduction, Page4, Lines 101-102: ‘Survivors of childhood cancer are also more likely to have other chronic health conditions’. Please provide examples of chronic conditions in survivors that can affect treatment.

Response: We have edited the introduction as suggested (please see below).

Edits to Introduction (edits are highlighted): Survivors of childhood cancer are also more likely to have other chronic health conditions¹² which may limit subsequent breast cancer treatments that can be delivered safely and also increase their susceptibility to treatment-related toxicities. For example, cardiovascular late effects including cardiomyopathy and heart failure¹³⁻¹⁶ are associated with increasing cumulative exposure to anthracyclines, agents that are also frequently used to treat breast cancer^{17,18} but are presumably avoided or limited when treating subsequent breast cancer.

22. Patients and Methods, Page 5: Lines 118-119: Please add a short paragraph about Childhood Cancer Survivor Study. Saying that 'details of the CCSS methodology are published elsewhere' without providing any information about the aim of the study and number of survivors recruited is not acceptable.

Response: We thank the reviewer for their recommendation. We have edited the manuscript to describe CCSS at first mention (located in the first line of the Results, because the journal requires the Methods section to follow Results).

Edits to Results (edits are highlighted): Female survivors of childhood cancer with subsequent breast cancer were identified in the Childhood Cancer Survivor Study (CCSS), a North American multi-institutional retrospective cohort study including 23,558 survivors and designed to quantify and understand the effects of pediatric cancer and treatment on long-term health.^{19,20} Among 11,550 females participating in CCSS, 431 had a pathology-

ascertained subsequent breast tumor (in situ or invasive) during adulthood (age ≥ 18 years) occurring at least five years after primary cancer diagnosis (Supplementary Figure 1).

23. Lines 126-131: The authors mentioned that neoplasms classified with a behaviour code 3 (i.e., invasive cancers) were considered subsequent malignant neoplasms. They also included in situ diseases. However, it is unclear why non-invasive breast cancers (such as DCIS and LCIS) were included. Excluding in situ cancers from survival analysis is generally recommended because they are non-invasive and typically have different prognoses and treatment approaches compared to invasive cancers. Including them might skew survival statistics.

Response: We thank the reviewer for raising this point. In situ subsequent breast cancers were included given previous evidence indicating their increased risk of mortality (Moskowitz *et al.*, Journal of Clinical Oncology, 2019). We also provided results for subgroups with LCIS/DCIS and invasive breast cancers separately (Figure 4, top middle and top right panels; provided below). Corresponding edits to the Discussion are included below.

Edits to Discussion (edits are highlighted): ...Including female survivors with in situ subsequent breast cancers affect estimates of excess mortality. These were included given previous evidence indicating their increased risk of mortality especially for non-breast cancer-related causes.⁵ We conducted analyses limited to and excluding those with in situ carcinomas; notably, we observed conditional excess mortality persists after excluding in situ carcinomas.

Figure 4: All-cause mortality probabilities in childhood cancer survivors and matched controls by breast cancer histology and treatment modality. In all panels, mortality curves for survivors are shown in blue while those for matched controls are shown in gray, and hazard ratios (HRs) adjusted for guideline-concordant breast cancer treatment comparing mortality risks for survivors and controls are provided along with corresponding 95% confidence intervals (CIs). Differences in mortality curves for survivors and matched controls were evaluated with two-sided log-rank tests with robust variance estimation; these p-values are shown in the lower left quadrant for each panel.

Statistical analysis

24. Lines 162-163: For mortality analyses, the index event was breast cancer diagnosis with time at risk ending at death or censoring for participants who were alive at the time of data abstraction. In the above sentence, please include the date of censoring and the date up to which the data on mortality was complete.

Response: We have included the last medical record abstraction follow-up date in the revised Methods (below), which is the date up to vital status/censoring was complete.

Edits to Methods (edits are highlighted): For mortality analyses, the index event was breast cancer diagnosis with time at risk ending at death or censoring for participants who were alive at the time of data abstraction (follow-up ending on December 28, 2022).

Discussion

25. The main limitation of this study is the unavailability of causes of death for the matched controls. Consequently, comparing mortality after breast cancer diagnosis between survivors and controls is not informative, as survivors have a higher prevalence of comorbidities due to previous cancer treatments. To determine whether breast cancer therapeutic modifications and treatment-related toxicities contribute to excess mortality in survivors, the authors should have examined cause-specific mortality, specifically deaths from breast cancer. I understand that this data was not available for controls, so this significant limitation should be thoroughly addressed in the discussion.

Response: We agree with the reviewer and have edited the Discussion to describe this limitation and contextualize our findings (please see below).

Edits to Discussion (edits are highlighted): Another important limitation of the present study is the lack of cause of death information among matched controls. We found breast cancer was the predominant cause of death among survivors with invasive tumors, including survivors diagnosed with early-stage (I/II) disease. While we would expect excess all-cause mortality to persist when comparing survivors and matched controls with invasive breast cancer given the greater contribution of non-breast cancer-related deaths among survivors, we observed considerably higher all-cause mortality probabilities at time points closer to breast cancer diagnosis among survivors with invasive versus in situ carcinomas (5-year: 16% versus 3%; 10-year: 33% versus 17%). We speculate that these findings may reflect excess subsequent breast cancer-related deaths among childhood cancer survivors despite therapeutic tradeoffs. This is consistent with previous work that reported a modest increase in breast cancer-related death among survivors, albeit without consideration of detailed breast cancer treatment data.⁵ Further study is needed to assess the magnitude of excess subsequent breast cancer-related death considering treatment modifications and acute toxicities, but requires cause of death and suitable breast cancer treatment data in a comparison group of females with first primary breast cancer.

26. If possible, please present the results (HRs) from Table 2 as Forest Plots.

Response: We thank the reviewer for their suggestion. We have edited Table 2 so that HRs and 95% CIs are presented as forest plots. Please see the updated Figure 2 below.

Figure 2: Risk factors for all-cause and cause-specific mortality after subsequent breast cancer in childhood cancer survivors. Abbreviations: BC, breast cancer; RT, radiation therapy; CV, cardiovascular; SMN, subsequent malignant neoplasm. HRs and corresponding 95% confidence intervals (CIs) from multivariable Cox proportional hazards regression models evaluating associations between primary and breast cancer treatments and the mortality hazard rate using age as the time scale and adjusting for covariates included above are shown, as well as breast cancer diagnosis year and histology (invasive versus non-invasive) or stage (III/IV versus I/II, invasive carcinomas only). HRs and 95% CIs are annotated for statistically significant variables (two-sided $P < 0.05$). Primary cancer chest RT dose is provided per 10 Gray and primary anthracycline dose is provided per 100 mg/m².

Other revisions

1. Corrections to Figure 4 (previously Figure 3 in the original manuscript submission): In the revision process, we found that we previously included an outdated version with incorrect 95% confidence intervals. This correction does not change any conclusions in our originally presented results. The specific changes in the manuscript main text are shown below in yellow highlights.

Relevant text in Results (changes highlighted): In analyses with matched controls with medical record- or registry-ascertained mortality data (n=241 pairs with 84 deaths among survivors and 33 deaths among controls), the 15-year mortality probability after breast cancer diagnosis was 41% among survivors (95% CI: 32-49%) versus 14% in controls (95% CI: 9-19%) (HR=3.48, 95% CI: 2.19-5.54; Figure 4, Supplementary Table 11). This excess mortality risk between survivors and the general population was substantial among those with in situ disease (HR=9.94, 95% CI: 3.13-31.57) and persisted among those with invasive disease (HR=2.88, 95% CI: 1.77-4.71).

References

1. Giaquinto, A.N., *et al.* Breast cancer statistics, 2022. *CA: a cancer journal for clinicians* **72**, 524-541 (2022).
2. Hill, D.A., Prossnitz, E.R., Royce, M. & Nibbe, A. Temporal trends in breast cancer survival by race and ethnicity: A population-based cohort study. *PLoS One* **14**, e0224064 (2019).
3. Lin, D.Y. & Wei, L.-J. The robust inference for the Cox proportional hazards model. *Journal of the American statistical Association* **84**, 1074-1078 (1989).
4. Kom, E.L., Graubard, B.I. & Midthune, D. Time-to-event analysis of longitudinal follow-up of a survey: choice of the time-scale. *Am J Epidemiol* **145**, 72-80 (1997).
5. Moskowitz, C.S., *et al.* Mortality After Breast Cancer Among Survivors of Childhood Cancer: A Report From the Childhood Cancer Survivor Study. *J Clin Oncol* **37**, 2120-2130 (2019).
6. Pfister, D.G., *et al.* Risk adjusting survival outcomes in hospitals that treat patients with cancer without information on cancer stage. *JAMA oncology* **1**, 1303-1310 (2015).
7. Gutierrez, J.C., *et al.* Are many community hospitals undertreating breast cancer?: lessons from 24,834 patients. *Annals of surgery* **248**, 154-162 (2008).
8. Lian, W., *et al.* The Impact of Young Age for Prognosis by Subtype in Women with Early Breast Cancer. *Scientific reports* **7**, 11625 (2017).
9. Kim, S.W., *et al.* Young Age Is Associated with Increased Locoregional Recurrence in Node-Positive Breast Cancer with Luminal Subtypes. *Cancer Res Treat* **49**, 484-493 (2017).
10. Hou, J., *et al.* Young age is associated with inferior outcomes in early-stage luminal B breast cancer patients who undergo mastectomy. *Future oncology (London, England)* **19**, 715-726 (2023).
11. Wang, Y.A., *et al.* Germline breast cancer susceptibility gene mutations and breast cancer outcomes. *BMC Cancer* **18**, 315 (2018).
12. Bhakta, N., *et al.* The cumulative burden of surviving childhood cancer: an initial report from the St Jude Lifetime Cohort Study (SJLIFE). *Lancet* **390**, 2569-2582 (2017).
13. Jasra, S. & Anampa, J. Anthracycline use for early stage breast cancer in the modern era: a review. *Current treatment options in oncology* **19**, 1-17 (2018).
14. Singal, P.K. & Iliskovic, N. Doxorubicin-induced cardiomyopathy. *New England Journal of Medicine* **339**, 900-905 (1998).
15. Qin, A., Thompson, C.L. & Silverman, P. Predictors of late-onset heart failure in breast cancer patients treated with doxorubicin. *Journal of Cancer Survivorship* **9**, 252-259 (2015).
16. Lipshultz, S.E., *et al.* Long-term cardiovascular toxicity in children, adolescents, and young adults who receive cancer therapy: pathophysiology, course, monitoring, management, prevention, and research directions: A scientific statement from the American Heart Association. *Circulation* **128**, 1927-1995 (2013).
17. Group, E.B.C.T.C. Comparisons between different polychemotherapy regimens for early breast cancer: meta-analyses of long-term outcome among 100 000 women in 123 randomised trials. *The Lancet* **379**, 432-444 (2012).
18. Waks, A.G. & Winer, E.P. Breast cancer treatment: a review. *Jama* **321**, 288-300 (2019).
19. Robison, L.L., *et al.* The Childhood Cancer Survivor Study: a National Cancer Institute–supported resource for outcome and intervention research. *Journal of clinical oncology* **27**, 2308 (2009).
20. Leisenring, W.M., *et al.* Pediatric cancer survivorship research: experience of the Childhood Cancer Survivor Study. *J Clin Oncol* **27**, 2319-2327 (2009).